# A bacterial immunity protein directly senses two disparate phage proteins

Tong Zhang[1], Albinas Cepauskas[2], Anastasiia Nadieina[2], Aurelien Thureau[3], Kyo Coppieters 't Wallant[4], Chloé Martens[4], Daniel C. Lim[1], Abel Garcia-Pino[2,5] ✉ & Michael T. Laub[1,6] ✉

Eukaryotic innate immune systems use pattern recognition receptors to sense infection by detecting pathogen-associated molecular patterns, which then triggers an immune response. Bacteria have similarly evolved immunity proteins that sense certain components of their viral predators, known as bacteriophages[1–6]. Although different immunity proteins can recognize different phage-encoded triggers, individual bacterial immunity proteins have been found to sense only a single trigger during infection, suggesting a one-to-one relationship between bacterial pattern recognition receptors and their ligands[7–11]. Here we demonstrate that the antiphage defence protein CapRel[SJ46] in *Escherichia coli* can directly bind and sense two completely unrelated and structurally different proteins using the same sensory domain, with overlapping but distinct interfaces. Our results highlight the notable versatility of an immune sensory domain, which may be a common property of antiphage defence systems that enables them to keep pace with their rapidly evolving viral predators. We found that Bas11 phages harbour both trigger proteins that are sensed by CapRel[SJ46] during infection, and we demonstrate that such phages can fully evade CapRel[SJ46] defence only when both triggers are mutated. Our work shows how a bacterial immune system that senses more than one trigger can help prevent phages from easily escaping detection, and it may allow the detection of a broader range of phages. More generally, our findings illustrate unexpected multifactorial sensing by bacterial defence systems and complex coevolutionary relationships between them and their phage-encoded triggers.

A central facet of innate immunity is the use of pattern recognition receptors (PRRs) that bind specific pathogen-associated molecular patterns (PAMPs), leading to the activation of cell-intrinsic defence mechanisms[12,13]. In mammals, diverse PRRs recognize different PAMPs. Canonically, mammalian PRRs are thought to be specific for a single PAMP—for example, RIG-I binds double-stranded RNA, TLR4 binds LPS and TLR5 binds flagellin[13]. Human NAIP/NLRC4 recognizes three different ligands—bacterial flagellin and the needle and inner rod proteins of the type III secretion system—but a common structural motif is recognized in each protein[14–16]. Other eukaryotic restriction factors are also often specific for individual viral proteins[17]. Host immunity proteins and the pathogen-encoded molecules they bind often engage in Red Queen coevolutionary dynamics[18]. This dynamic is typically framed as a molecular arms race in which a single protein being sensed can acquire mutations to evade detection, leading to selective pressure on the host factor to mutate and restore the interaction.

The concept of PRRs and PAMPs extends to bacteria and their ability to detect infection by bacteriophages. Recent work indicates that bacteria harbour proteins analogous to PRRs that recognize certain phage proteins or nucleic acids during an infection, leading to the activation of various antiphage defence mechanisms[7–11,19–21]. There are only a handful of cases for which the direct trigger of an antiphage defence system is known, so the specificity of phage detection by bacterial PRRs is largely unknown. Individual defence systems have been reported to recognize only single ligands during phage infection. In addition, a screen for phages that escape various defence systems, an approach that can unveil phage-encoded PAMPs, primarily identified mutations in a single phage gene for each defence system examined[7]. These previous results suggest that bacterial PRRs, like most eukaryotic PRRs, also typically have one-to-one relationships with their phage-encoded triggers. However, the homologues of a given family of bacterial PRRs can sometimes recognize different PAMPs[8], and one large-scale screen indicated that some bacterial defence proteins can be activated by the ectopic expression of multiple phage proteins[19]. Here we demonstrate that the antiphage defence protein, CapRel[SJ46], directly binds and senses two completely unrelated and structurally different phage proteins using

[1]Department of Biology, Massachusetts Institute of Technology, Cambridge, MA, USA. [2]Cellular and Molecular Microbiology, Faculté des Sciences, Université Libre de Bruxelles (ULB), Brussels, Belgium. [3]Centre for Structural Biology and Bioinformatics, Université Libre de Bruxelles (ULB), Brussels, Belgium. [4]Synchrotron SOLEIL, Gif sur Yvette, France. [5]WELBIO, Brussels, Belgium. [6]Howard Hughes Medical Institute, Massachusetts Institute of Technology, Cambridge, MA, USA. ✉e-mail: abel.garcia.pino@ulb.be; laub@mit.edu

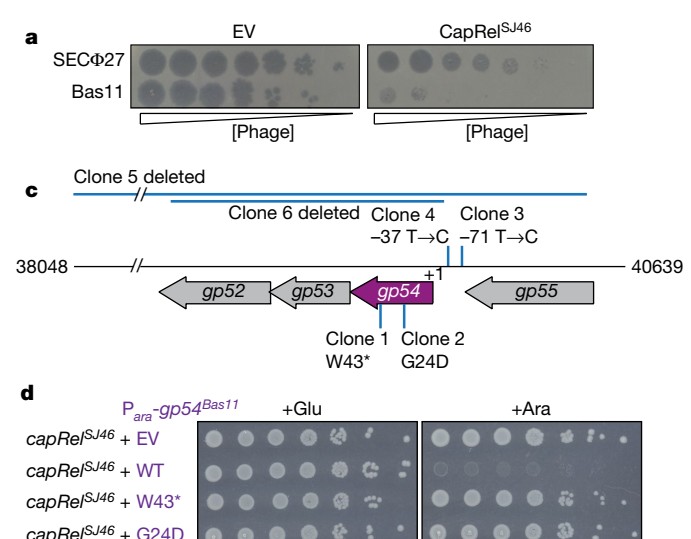

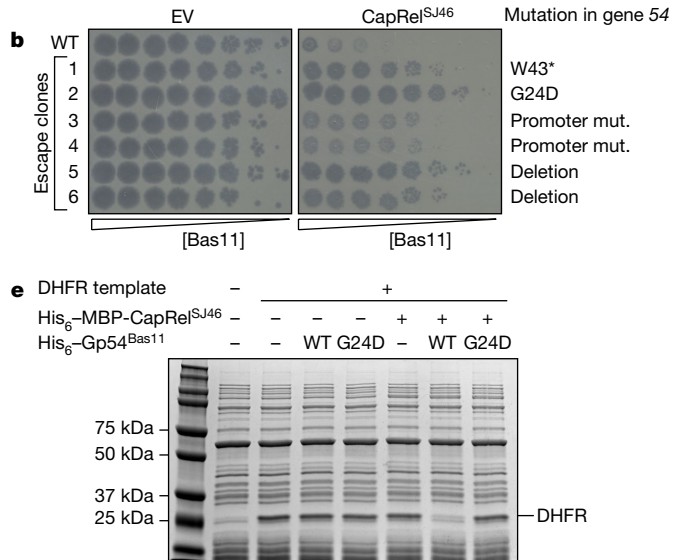

**Fig. 1 | Gp54 in Bas11 is an activator of the CapRel^SJ46 defence system.**
**a**, Serial, tenfold dilutions of the indicated phages spotted on lawns of cells harbouring an empty vector (EV) or plasmid producing CapRel^SJ46. Relative phage concentration is indicated by the height of the wedge. **b**, Serial dilutions of six escape clones of Bas11 and a control wild-type (WT) phage spotted on lawns of cells harbouring either an empty vector or a CapRel^SJ46 expression vector, with the corresponding mutations in gene *54* labelled. **c**, Schematic of the gene *54* genomic region in Bas11, with mutations in the escape clones

from **b** labelled. **d**, Cell viability assessed by serial dilutions of cells producing CapRel^SJ46 from its native promoter, and the indicated variant of Gp54^Bas11 from an arabinose-inducible promoter (P_ara) on medium containing glucose (Glu) or arabinose (Ara). **e**, In vitro transcription–translation assays using DHFR production from a DNA template as readout. Purified hexahistidine (His_6)- and maltose binding protein (MBP)-tagged CapRel^SJ46 (His_6–MBP–CapRel^SJ46) and either the wild-type or G24D variant of His_6–Gp54^Bas11 were added to the reactions. Image shown is representative of three biological replicates.

the same sensor domain. We find one phage that harbours both trigger proteins and demonstrate that they both activate CapRel^SJ46 during infection. Our results indicate that the Red Queen dynamic unfolding between bacteria and phages may not always involve a single PRR and a single ligand, but rather involve the complex coevolution of multiple factors.

## Gp54 is an alternative trigger of CapRel

We recently identified and characterized a fused toxin–antitoxin system called CapRel^SJ46 that protects *Escherichia coli* against diverse phages[11]. CapRel^SJ46 contains an N-terminal toxin domain and a C-terminal antitoxin domain that normally binds and autoinhibits the N-terminal toxin. During infection by SECΦ27 phage, the newly synthesized major capsid protein (MCP) binds to the C-terminal domain of CapRel^SJ46 to relieve autoinhibition, leading to the activation of CapRel^SJ46. Activated CapRel^SJ46 then pyrophosphorylates the 3′ end of transfer RNAs, which inhibits protein translation and restricts phage propagation[11]. Sensing the MCP, which is an essential and abundant component of the phage, is beneficial to the host bacteria because it limits the number of mutations that phages can acquire to escape defence. However, given intense selective pressure to maintain infectivity, phages can evolve to overcome defence through mutations in their capsid protein. For instance, a SECΦ27-like phage called Bas4 naturally encodes a single amino acid substitution in its MCP that prevents activation of CapRel^SJ46, enabling the phage to escape defence[11]. Such escape may drive selection for mutations in CapRel^SJ46 that restore an interaction with the MCP. Alternatively, CapRel^SJ46 could, in principle, evolve to sense a different phage protein.

To explore whether CapRel^SJ46 can sense phage factors other than the MCP, we focused on a family of phages from the BASEL collection[22] that are closely related to SECΦ27, including Bas11. When CapRel^SJ46 was produced from its native promoter on a low-copy-number plasmid in *E. coli* MG1655, it decreased the efficiency of plaquing (EOP) of Bas11 by over 10^4-fold (Fig. 1a and Extended Data Fig. 1a), indicating

that it had provided strong defence against Bas11. To identify the phage-encoded activator(s) in Bas11, we isolated spontaneous Bas11 mutants that largely overcome CapRel^SJ46 defence (Fig. 1b and Extended Data Fig. 1b). Notably, although the MCP of Bas11 is highly similar (85% identical) to that of SECΦ27 (Extended Data Fig. 1c), none of the Bas11 escape mutants mapped to its MCP. Instead, all six escaping phage clones contained mutations in the genomic region of gene *54*, which encodes a small hypothetical protein of 66 amino acids, Gp54^Bas11 (Fig. 1b,c). Escape clones 1 and 2 each had a single nucleotide substitution that led to either a premature stop codon (W43*) or a single amino acid substitution (G24D) in Gp54^Bas11. Clones 3 and 4 each contained a mutation immediately upstream of the gene *54* coding region, probably within its promoter (Fig. 1c). Expression levels of Gp54^Bas11 from the mutated promoters were lower than that from wild-type promoter (Extended Data Fig. 1d). Lastly, clones 5 and 6 each had a large deletion encompassing gene *54* and nearby genes (Fig. 1c). These results suggested that loss-of-function mutations in gene *54* allow Bas11 to largely overcome CapRel^SJ46 defence. Notably, Gp54^Bas11 does not show any sequence similarity to the MCP of SECΦ27.

We hypothesized that the wild-type phage protein Gp54^Bas11 may be an activator of CapRel^SJ46, with the escape mutants enabling the phage overcome defence by preventing activation. To test whether Gp54^Bas11 is sufficient to activate CapRel^SJ46, which blocks cell growth when active[11], we coproduced either wild-type or a mutant variant of Gp54^Bas11 with CapRel^SJ46 in the absence of phage infection. Wild-type Gp54^Bas11 rendered CapRel^SJ46 toxic, whereas neither variant (W43* or G24D) had any effect on cell growth when coproduced with CapRel^SJ46 (Fig. 1d). As a control, we verified that neither wild-type nor mutant variants of Gp54^Bas11 were toxic in the absence of CapRel^SJ46 (Extended Data Fig. 1e).

We then tested whether Gp54^Bas11 can activate CapRel^SJ46 to inhibit protein translation in a reconstituted in vitro transcription–translation system. Incubation of purified His_6–MBP–CapRel^SJ46 with purified His_6–Gp54^Bas11 strongly inhibited the synthesis of a model protein, DHFR, whereas the G24D variant of Gp54^Bas11 had no effect (Fig. 1e). We verified that the G24D variant was still properly folded because it had

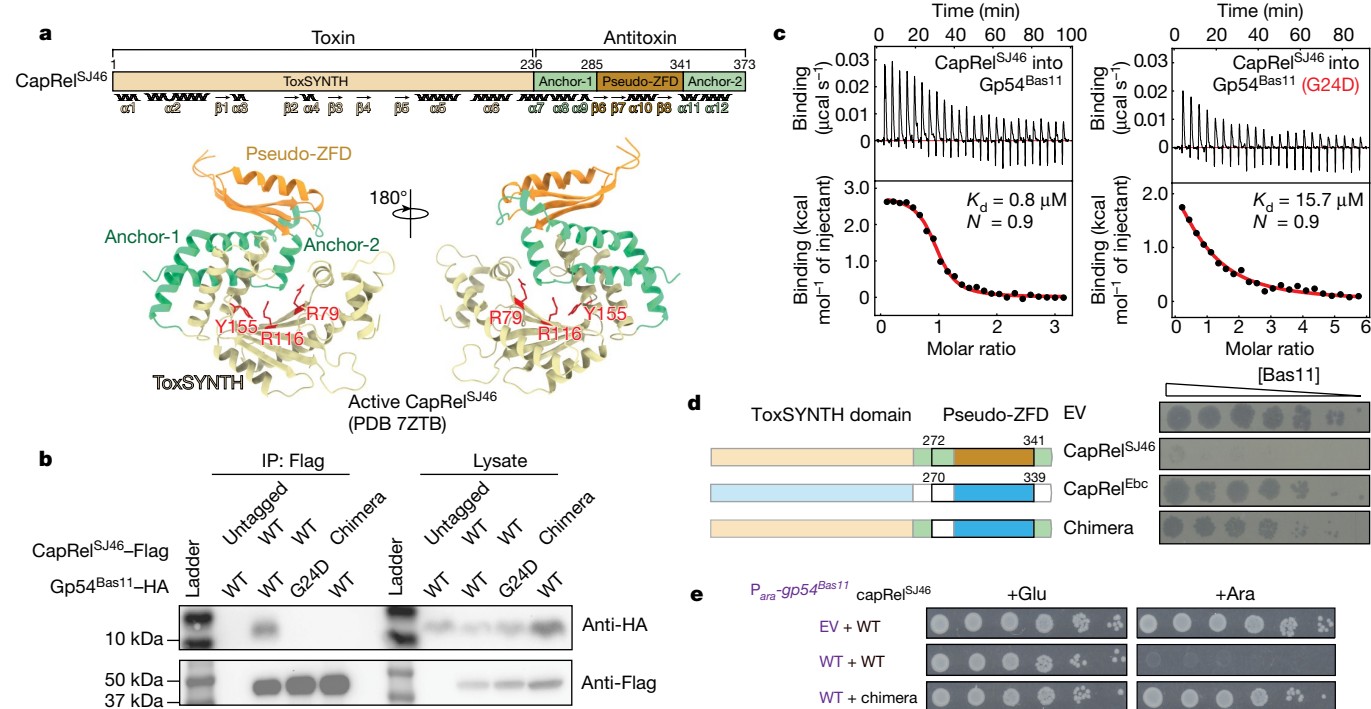

**Fig. 2 | Gp54$^{Bas11}$ binds directly to the antitoxin region of CapRel$^{SJ46}$.**
**a**, Schematic of the domain organization of CapRel$^{SJ46}$ (top) and cartoon representation of the crystal structure of CapRel$^{SJ46}$ coloured by domains (bottom). Active site G-loop Y155 and ATP-coordination residues R79 and R116 of the toxin domain (toxSYNTH) are highlighted in red. **b**, Flag-tagged CapRel$^{SJ46}$ (CapRel$^{SJ46}$–Flag) or chimera–Flag was immunoprecipitated from cells producing CapRel$^{SJ46}$–Flag or chimera–Flag and haemagglutinin (HA)-tagged Gp54$^{Bas11}$ (wild type or the G24D variant), and probed for the presence of the indicated Gp54$^{Bas11}$ variant using the HA tag. Lysates used as input for immunoprecipitation (IP) were probed as controls for expression levels. Image shown is representative of two biological replicates. **c**, Binding of CapRel$^{SJ46}$ to the wild-type or G24D variant of Gp54$^{Bas11}$ monitored by ITC, with binding affinity ($K_d$) and stoichiometry ($N$) noted. **d**, Left, schematic of the CapRel constructs. Right, serial dilutions of the Bas11 phage spotted on lawns of cells harbouring either the indicated CapRel constructs or an empty vector. **e**, Serial dilutions of cells producing CapRel$^{SJ46}$ or the chimera from its native promoter and wild-type Gp54$^{Bas11}$ from an arabinose-inducible promoter on medium containing glucose or arabinose.

a circular dichroism spectrum comparable to the wild-type protein (Extended Data Fig. 1f). Together, our results indicated that wild-type Gp54$^{Bas11}$, like the previously identified MCP from phage SECΦ27, activates CapRel$^{SJ46}$.

## CapRel antitoxin directly senses Gp54

CapRel$^{SJ46}$ consists of a conserved N-terminal toxin domain that can pyrophosphorylate tRNAs and a C-terminal antitoxin domain containing a zinc-finger-like domain (pseudo-ZFD), flanked by α-helices referred to as anchors (Fig. 2a). The antitoxin domain is highly variable among CapRel homologues (Extended Data Fig. 2a) and largely determines the phage specificity of CapRel defence[11]. The antitoxin of CapRel$^{SJ46}$ directly binds the MCP of SECΦ27, serving as a phage infection sensor[11]. To test whether Gp54$^{Bas11}$ also interacts with CapRel$^{SJ46}$ to activate it, we immunoprecipitated CapRel$^{SJ46}$–Flag from cells coproducing wild-type Gp54$^{Bas11}$–HA or the G24D variant, having verified that tags did not affect protein functions (Extended Data Fig. 2b). We found that wild-type, but not the G24D variant of, Gp54$^{Bas11}$ coimmunoprecipitated with CapRel$^{SJ46}$ (Fig. 2b). In addition, isothermal titration calorimetry (ITC) indicated that purified Gp54$^{Bas11}$ directly binds CapRel$^{SJ46}$ with a $K_d$ of 800 nM in a 1:1 ratio (Fig. 2c), comparable to that previously measured for MCP$^{SECΦ27}$ (350 nM)[11]. The interaction is entropically driven, suggesting either that the bound state is somewhat dynamic or that some region of the complex becomes disordered following binding. Binding affinity decreased at least 20-fold for the G24D variant of Gp54$^{Bas11}$ (Fig. 2c).

To test whether Gp54$^{Bas11}$ is also sensed by the antitoxin domain of CapRel$^{SJ46}$, we spotted Bas11 phages onto cells producing the homologue CapRel$^{Ebc}$ from *Enterobacter chengduensis* or a chimera that replaced most of the CapRel$^{SJ46}$ antitoxin with the corresponding region of CapRel$^{Ebc}$ (Fig. 2d and Extended Data Fig. 2c). Unlike CapRel$^{SJ46}$, neither CapRel$^{Ebc}$ nor the chimera provided robust defence against Bas11 (Fig. 2d and Extended Data Fig. 2c), despite their abilities to protect against another phage, T7 (Extended Data Fig. 2d). In addition, the chimeric version of CapRel was no longer toxic to cells when coproduced with wild-type Gp54$^{Bas11}$ (Fig. 2e), and Gp54$^{Bas11}$ did not coprecipitate with the chimeric CapRel (Fig. 2b). These results indicated that the antitoxin domain of CapRel$^{SJ46}$ is important for sensing Gp54$^{Bas11}$, despite the lack of sequence similarity of Gp54 to the MCP from SECΦ27.

To further investigate how the antitoxin of CapRel$^{SJ46}$ senses Gp54$^{Bas11}$, we mutagenized this domain through error-prone PCR and selected for CapRel$^{SJ46}$ mutants that were no longer activated by Gp54$^{Bas11}$. The single substitutions N275D (Fig. 3a), L270P and L276P (Extended Data Fig. 3a) each largely abolished the toxicity of CapRel$^{SJ46}$ when coproduced with Gp54$^{Bas11}$, and substantially weakened CapRel$^{SJ46}$ defence against Bas11 (Extended Data Fig. 3b). Notably, these residues all lie within α-helix 9, which is part of anchor-1 in the antitoxin and is highly variable among CapRel homologues (Fig. 2a and Extended Data Figs. 2a and 3c); thus, we hypothesized that α9, formed by residues 270–279, might be critical for the interaction of CapRel$^{SJ46}$ with Gp54$^{Bas11}$. To further probe the role of anchor-1 in interacting with Gp54$^{Bas11}$, we made substitutions in other non-conserved residues within α9 and tested their activation by Gp54$^{Bas11}$. The substitutions K278E (Fig. 3a), D273K and S279P (Extended Data Fig. 3a) each reduced or abolished the toxicity of CapRel$^{SJ46}$ following induction of Gp54$^{Bas11}$, supporting a key role for this helix in sensing Gp54$^{Bas11}$.

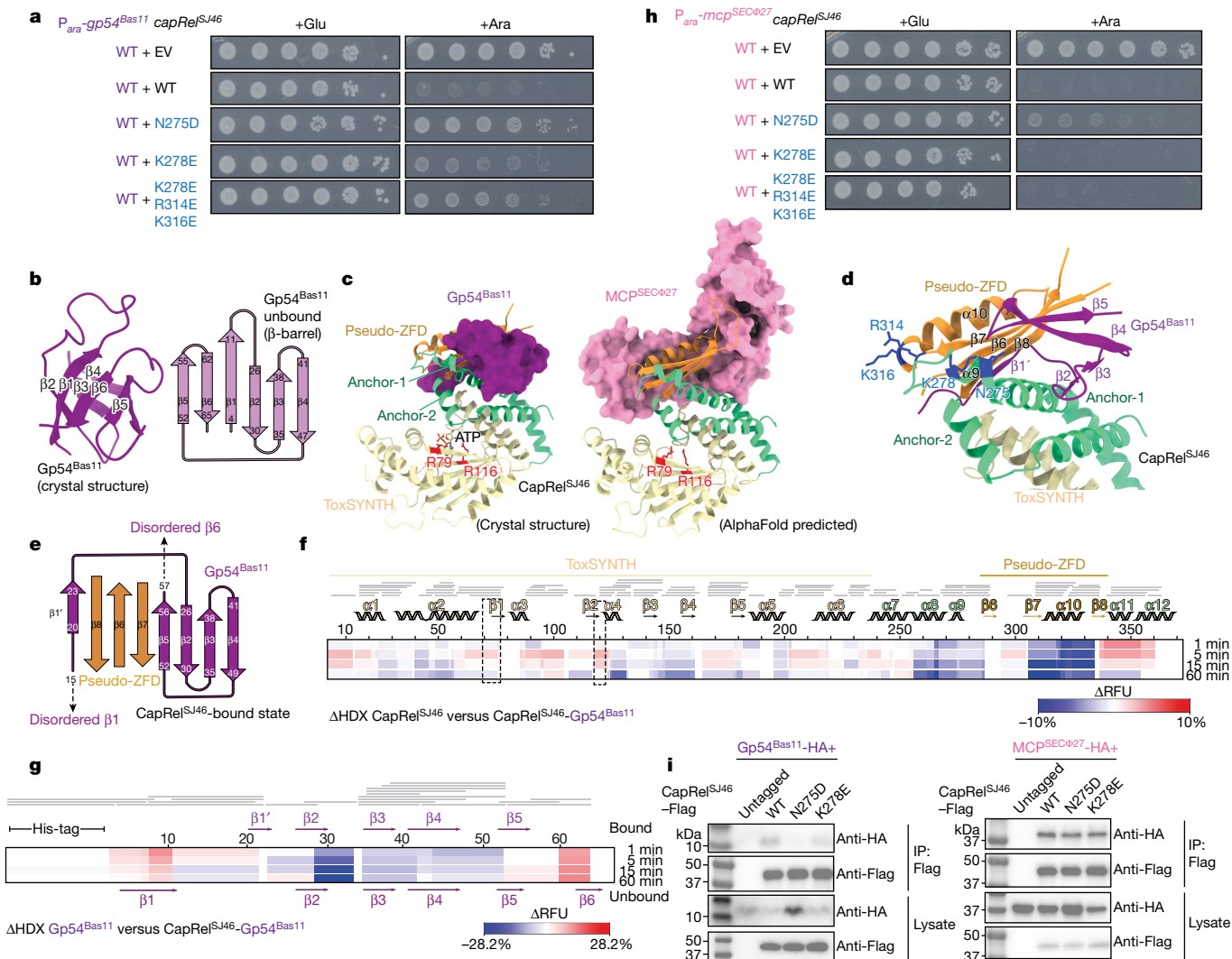

**Fig. 3 | Gp54[Bas11] and MCP[SECΦ27] bind overlapping but distinct regions of CapRel[SJ46]. a**, Serial dilutions of cells producing the indicated variant of CapRel[SJ46] from its native promoter and wild-type Gp54[Bas11] from an arabinose-inducible promoter on medium containing glucose or arabinose. **b**, Left, cartoon representation of the crystal structure of Gp54[Bas11]. Right, topological representation of Gp54[Bas11] in an unbound, β-barrel state. **c**, Left, crystal structure of the complex of Gp54[Bas11] (purple) bound to CapRel[SJ46] (coloured by domains). Right, predicted structural model of the complex of CapRel[SJ46] and MCP[SECΦ27] (pink) by AlphaFold. ATP-coordination residues of the CapRel[SJ46] toxin domain are highlighted in red. **d**, Details of the interface formed by the antitoxin domain of CapRel[SJ46] and Gp54[Bas11] (purple), with the residues substituted coloured in blue. **e**, Topological representation of Gp54[Bas11] (purple) in a CapRel[SJ46]-bound state involving interaction with the pseudo-ZFD of

CapRel[SJ46] (orange). **f**, Differential HDX (ΔHDX) between CapRel[SJ46] and CapRel[SJ46]–Gp54[Bas11] shown as a differential heat map. Change in relative fractional units (ΔRFU) is colour coded, with red indicating increased deuteration of CapRel[SJ46] in the presence of Gp54[Bas11] and blue indicating lower deuteration. Grey bars indicate peptides identified in mass spectrometry analysis. Regions corresponding to the toxSYNTH active site highlighted by dashed-line boxes. **g**, As in **f** but comparing Gp54[Bas11] and CapRel[SJ46]–Gp54[Bas11]. **h**, As in **a** but with the wild-type MCP from SECΦ27. **i**, CapRel[SJ46]–Flag or the indicated variant was immunoprecipitated from cells producing CapRel[SJ46]–Flag and Gp54[Bas11]–HA or MCP[SECΦ27]–HA, and probed for the presence of Gp54[Bas11] or MCP[SECΦ27] using the HA tag. Image shown is representative of two biological replicates.

## Gp54 changes conformation to bind CapRel

To gain better structural insight into the interaction between CapRel[SJ46] and Gp54[Bas11], we first solved a crystal structure of Gp54[Bas11] to 2.3 Å resolution (Fig. 3b and Extended Data Table 1). This structure showed a small, six-stranded β-barrel with one prominent loop between β-strands β1 and β2; β-barrels with this topology are very rare in nature[23]. The closest structural homologues of Gp54[Bas11] are five-stranded β-barrel SH3 domains (DALI z-score of 3.8), with a 3₁₀ α-helix replacing the additional β-strand (Extended Data Fig. 3d). Thus, the structure of Gp54[Bas11] is significantly distinct from the AlphaFold-predicted structure of MCP from SECΦ27, with root mean square deviation (r.m.s.d.) greater than 14 Å (Extended Data Fig. 3e).

Next, we determined the structure of a CapRel[SJ46]–Gp54[Bas11] complex to 2.2 Å resolution (Fig. 3c, Extended Data Fig. 4a–c and Extended Data Table 1). This complex had ATP bound in the pyrophosphate donor site (Fig. 3c and Extended Data Fig. 4c), stacked between R79 and R116, similar to that of other RelA/SpoT homologue enzymes[24,25], which indicated that the complex captures the active state of the enzyme. The complex structure showed that Gp54[Bas11] interacts with the pseudo-ZFD and anchor-1 of the antitoxin, with an interface of about 1,650 Å² that is partially overlapping, but largely distinct from, that formed between CapRel[SJ46] and MCP[SECΦ27] as predicted by AlphaFold and previously validated[11] (Fig. 3c and Extended Data Fig. 4d). In this triggered state, residue Y355 in CapRel[SJ46], which is part of the YXXY motif that normally blocks the ATP-binding site in the closed state[11],

is tethered to Gp54[Bas11] β2 by K269 and cannot interact with the toxin domain (Extended Data Fig. 4e). Small-angle X-ray scattering (SAXS) analysis of the CapRel[SJ46]–Gp54[Bas11] complex was compatible with the crystal structure and indicated, by comparison with the unbound CapRel[SJ46], that Gp54[Bas11] effectively clamps the pseudo-ZFD to both anchors, precluding recoil toward the toxin active site (Extended Data Fig. 4f,g and Extended Data Table 2).

Each of the substitutions in CapRel[SJ46] identified above as affecting activation (Fig. 3a and Extended Data Fig. 3a,b) maps to the interface formed with Gp54[Bas11] (Fig. 3d and Extended Data Fig. 5a). In the complex, the hydrophobic core of the Gp54[Bas11] β-barrel binds to both the amphipathic anchor-1 and pseudo-ZFD of CapRel[SJ46]. In particular, G24, I25, S39, L41 and W43 of Gp54[Bas11] contact the pseudo-ZFD β-sheet, and W53 of Gp54[Bas11] becomes embedded between L270 and L276 of anchor-1 in CapRel[SJ46] and I29 and L31 of Gp54[Bas11] (Extended Data Fig. 5a,b). The interface is further stabilized by a polar network between D273, N275, K278 and S279 from CapRel[SJ46] and Q47 and N50 from Gp54[Bas11] (Extended Data Fig. 5a,b).

Notably, binding of Gp54[Bas11] to CapRel[SJ46] involves a significant topological rearrangement (Fig. 3b,e). In the complex, the β-barrel of Gp54[Bas11] unfolds with β1 and β6 becoming disordered, consistent with the entropy-driven binding suggested by ITC (Fig. 2c). Whereas Gp54[Bas11] β2–β5 bind on one side of the pseudo-ZFD of CapRel[SJ46] interacting with β7 and anchor-1, the long β1–β2 loop of Gp54[Bas11] folds into β-strand β1′ and binds on the other side, making contacts with β8 of the pseudo-ZFD and effectively clamping the pseudo-ZFD (Fig. 3d,e and Extended Data Fig. 4a). These interactions produce a 'hybrid' eight-stranded, antiparallel, twisted β-sheet comprising β-strands from both Gp54[Bas11] and CapRel[SJ46] that wraps around anchor-1 of CapRel[SJ46] (Fig. 3d,e and Extended Data Fig. 4a). In addition, binding by β1′ of Gp54[Bas11] to CapRel[SJ46] β8 moves the pseudo-ZFD further from the active site compared with the unbound open state of CapRel[SJ46], which probably primes the enzyme to bind and modify target tRNAs (Extended Data Fig. 5c). Finally, we noted that the N-terminal region of Gp54[Bas11] β1′ (and possibly the disordered N terminus) might also make contact with the cap of α10 (residues 314–317) in CapRel[SJ46] (Extended Data Fig. 5d). Supporting the relevance of this interaction, we found that substitutions R314E and K316E, together with K278E, further reduced the toxicity of CapRel[SJ46] following induction of Gp54[Bas11] (Fig. 3a).

To further validate the ordered-to-disordered transition of Gp54[Bas11] and binding interface of the CapRel[SJ46]–Gp54[Bas11] complex, we used hydrogen–deuterium exchange (HDX) coupled with mass spectrometry for comparison of the CapRel[SJ46]–Gp54[Bas11] complex with the unbound proteins. This analysis showed protection of anchor-1 and the pseudo-ZFD regions of CapRel[SJ46], particularly of β7–β8 and α8–α10 (Fig. 3f and Extended Data Fig. 6a–d), which almost perfectly matched the crystallographic interface. We also observed an increase in deuterium uptake by anchor-2 (α11 and α12), which contains the YXXY motif that interacts with the toxin active site (Fig. 3f and Extended Data Fig. 6d). This deprotection reflects the opening of CapRel[SJ46] following binding of Gp54[Bas11]. On the Gp54[Bas11] side, both the newly formed β1′ and the β2–β4 region showed decreases in deuterium uptake, consistent with the crystal structure (Fig. 3g). HDX mass spectrometry (HDX–MS) also showed an increase in uptake in the N- and C-terminal regions of Gp54[Bas11] (β1 and β6 in the unbound state), consistent with unfolding of the β-barrel following binding of CapRel[SJ46] and disorder in these regions in the crystal structure. Collectively, our results demonstrate that a dynamic, entropically favourable bound state drives the activation of CapRel[SJ46] by Gp54[Bas11].

## Genetic separation of CapRel activation

Importantly, the α-helix (α9) in anchor-1 of CapRel[SJ46] that makes extensive contacts with Gp54[Bas11] does not contribute significantly to the AlphaFold-predicted interface with MCP[SECΦ27] (Fig. 3c). Thus, we

hypothesized that the substitutions in this region of CapRel[SJ46] that disrupt activation by Gp54[Bas11] would not impact activation by MCP[SECΦ27]. To test this hypothesis, we coproduced our CapRel[SJ46] variants with MCP[SECΦ27] and found that the single substitutions N275D and K278E, as well as the triple substitution K278E/R314E/K316E, did not substantially affect activation by MCP[SECΦ27] despite their reduced activation by Gp54[Bas11] (Fig. 3h). We also found that the N275D and K278E variants of CapRel[SJ46] coprecipitated with MCP[SECΦ27] as well as the wild-type CapRel[SJ46], but, as expected based on the crystal structure (Fig. 3c,d), had reduced binding to Gp54[Bas11] in this assay (Fig. 3i).

Together, our results indicated that the antitoxin domain of CapRel[SJ46] is critical for sensing both the MCP of SECΦ27 and Gp54 from Bas11, with overlapping but not identical regions of the antitoxin involved in the two different interactions. More broadly, these findings show the notable versatility of a zinc-finger-like domain in recognizing different proteins, which enables sensing by a single bacterial defence protein of multiple phage-encoded activators.

## Other Gp54 homologues do not trigger CapRel

Given that both the MCP of SECΦ27 and Gp54 from Bas11 can activate CapRel[SJ46], we decided to examine a set of phages from the BASEL collection—Bas5, Bas8 and Bas10—that are closely related to SECΦ27 and Bas11 and encode homologues of both the MCP (Extended Data Fig. 1c) and Gp54 (Fig. 4a and Extended Data Fig. 7a). The region of the MCP demonstrated in SECΦ27 to mediate an interaction with CapRel[SJ46] was nearly identical in each of these phages (Extended Data Fig. 1c). When examining the Gp54 homologues, we noted that those from Bas11 and Bas10 were nearly identical, with only three amino acid differences, whereas the SECΦ27, Bas5 and Bas8 homologues contained more substitutions relative to Bas11 (Fig. 4a).

We previously showed that CapRel[SJ46] can defend against Bas5 and Bas8, like SECΦ27, by sensing their MCPs[11]. The MCPs of Bas5 and Bas8 are sufficient to activate CapRel[SJ46] on their own, and mutations in the MCPs allowed the phages to escape defence[11]. Here we found that CapRel[SJ46] also defends against Bas10, reducing plaquing by around tenfold, and selection for complete escape led to the identification of clones producing a single amino acid substitution (I85T or I115F) in its MCP, MCP[Bas10] (Fig. 4b and Extended Data Fig. 7b). In addition, we found that wild-type MCP from Bas10, but not the I115F variant, caused toxicity in cells coproducing CapRel[SJ46] (Fig. 4c). Notably, the same I115F substitution emerged when phage SECΦ27 was evolved to overcome CapRel[SJ46] defence[11].

Our results indicated that the MCP of Bas10, like that of Bas5, Bas8 and SECΦ27, is necessary and sufficient to activate CapRel[SJ46]. However, these four phages also harbour homologues of the unrelated, alternative activator Gp54 from Bas11. We therefore tested whether the Gp54 homologues of these phages can also activate CapRel[SJ46]. However, none of the Gp54 homologues from SECΦ27, Bas5, Bas8 and Bas10 caused cellular toxicity when coproduced with CapRel[SJ46] (Fig. 4d). These findings are consistent with our results demonstrating that mutations in the MCP-encoding gene of these phages enabled complete escape from CapRel[SJ46] defence. We concluded that phages SECΦ27, Bas5, Bas8 and Bas10 contain only a single protein activator of CapRel[SJ46] (their MCPs), despite encoding homologues of the Gp54 activator found in Bas11 phage.

This conclusion was most unexpected in regard to Bas10, which is the most closely related to Bas11 and encodes a homologue of Gp54[Bas11] containing only three amino acid differences (Fig. 4a). To test the importance of these three residues for activation of CapRel[SJ46] by Gp54[Bas11], we made three single substitutions (D6E, A7V or I25V) in Gp54[Bas11] to individually introduce the residues found at these positions in the Bas10 homologue. The substitutions A7V and I25V in Gp54[Bas11] each largely abolished its ability to activate CapRel[SJ46] when coproduced, whereas D6E had no effect (Fig. 4e). We also made the reciprocal, individual

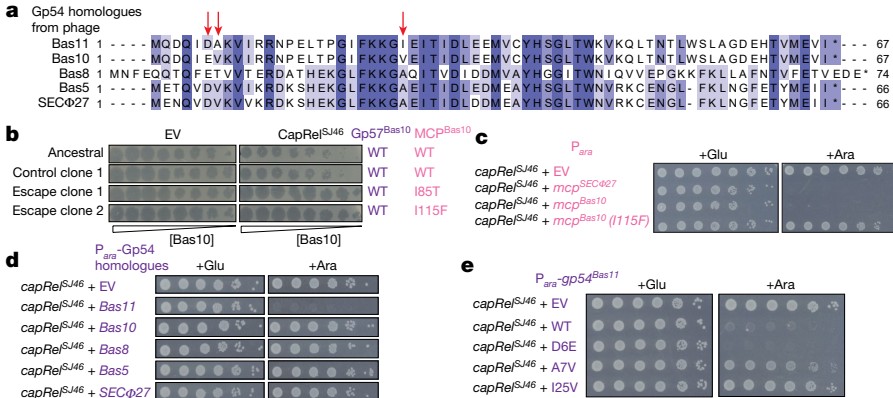

**Fig. 4 | Homologues of Gp54^Bas11 do not trigger CapRel^SJ46. a**, Multiple sequence alignment of Gp54 homologues from the indicated phages. Residues that are different between the Gp54 homologue from phage Bas11 and Bas10 are labelled by red arrows. **b**, Serial dilutions of wild-type (ancestral or from a control population evolved without selective pressure) and the escaping clones of Bas10 spotted on lawns of cells harbouring either an EV or a plasmid producing CapRel^SJ46. The corresponding genotypes of its MCP and its Gp54 homologue (Gp57^Bas10) are indicated. **c**, Serial dilutions of cells producing CapRel^SJ46 from its native promoter and the indicated MCP or its variant from an arabinose-inducible promoter on medium containing glucose or arabinose. **d**, As in **c** but with each Gp54 homologue from the indicated phages. **e**, As in **c** but with the indicated variant of Gp54^Bas11.

substitutions (V7A or V25I) in the Bas10 homologue of Gp54 but found that neither substitution alone enabled activation of CapRel^SJ46, whereas the double mutant was sufficient to activate (Extended Data Fig. 7c). I25 is found in the CapRel^SJ46–Gp54^Bas11 complex interface, as is the adjacent G24 (Extended Data Figs. 7d and 5b). The substitutions I25V and G24D (identified as an escape mutant) probably disrupt binding and thereby abolish activation. By contrast, A7 is disordered in the complex but is part of the β-barrel core in unbound Gp54^Bas11 (Extended Data Fig. 7e). The A7V substitution may have stabilized the unbound β-barrel, which would also have prevented activation by preventing the unfolding of Gp54^Bas11.

## Bas11 encodes two triggers of CapRel

The results presented thus far raised a conundrum: our escape mutant screen with Bas11 showed mutations only in gene *54* (Fig. 1b) and Gp54^Bas11 was sufficient to activate CapRel^SJ46 (Fig. 1d), but the MCP of Bas11 is identical to that of Bas10 in which the MCP is the sole trigger for CapRel^SJ46. We revisited our Bas11 escape mutants and noted that each only partially escaped CapRel^SJ46 defence, with reduction of about 10- to 100-fold in EOP (Fig. 1b and Extended Data Fig. 1b). Even the clone of Bas11 with a deletion of gene *54* (clone 5 in Fig. 1b) still formed smaller plaques when spotted onto cells containing CapRel^SJ46 compared with cells with an empty vector (Fig. 1b). We tried to evolve this mutant clone of Bas11 to completely overcome CapRel^SJ46 defence, and succeeded in isolating mutants that plaqued the same on CapRel^SJ46-containing cells as empty vector cells (Fig. 5a and Extended Data Fig. 8a). Notably, whole-phage genome sequencing showed that all of our escape mutants produced an I115F substitution in the MCP, MCP^Bas11, in addition to the deletion of the gene *54* region (Fig. 5a). As shown above, the wild-type MCP from Bas11 (which is identical to that of Bas10) was sufficient to activate CapRel^SJ46 and the substitution I115F completely ablated this activation (Fig. 4c). We engineered wild-type Bas11 phage to encode only the I115F substitution in its MCP—that is, with gene *54* present—and observed that this substitution alone was also insufficient for Bas11 to completely escape CapRel^SJ46 defence with a tenfold reduction in EOP and smaller plaques (Fig. 5b and Extended Data Fig. 8b). Thus, our results demonstrate that Bas11 encodes two activators of CapRel^SJ46 and that it can fully escape CapRel^SJ46 defence only when both activators are mutated.

To further compare the activation of CapRel^SJ46 by the MCP and Gp54^Bas11, we engineered phage SECΦ27 such that it encodes one or both proteins as activators. As shown previously, despite encoding a Gp54^Bas11 homologue (known as Gp19), wild-type SECΦ27 normally triggers CapRel^SJ46 only by its MCP, with a single substitution in the MCP (L114P) allowing the phage to completely escape defence. We replaced the coding sequence of the SECΦ27 homologue (Gp19) with that of Gp54^Bas11 in both wild-type SECΦ27 and the SECΦ27 MCP(L114P) escape phage (Fig. 5c). When Gp54^Bas11 was introduced into SECΦ27 MCP(L114P) it restored CapRel^SJ46-dependent defence, with a decrease of over 10^5-fold in EOP (Fig. 5c and Extended Data Fig. 8c). Defence against this phage, which uses Gp54^Bas11 as the activator of CapRel^SJ46, was stronger than that against wild-type SECΦ27, which produces approximately 10^2-fold reduction in EOP and uses the MCP only to trigger CapRel^SJ46 (Fig. 5c and Extended Data Fig. 8c). However, we found that some clones of this engineered phage spontaneously escaped CapRel^SJ46 defence (Fig. 5c). We isolated eight such clones that completely overcame defence and found that each harboured a mutation in the region encoding its activator Gp54^Bas11 (Fig. 5d,e and Extended Data Fig. 8d). Three clones had a single amino acid substitution (D6G, S39C or L41P) in Gp54^Bas11, and these variants no longer activated CapRel^SJ46 (Extended Data Fig. 8e). Notably, S39 and L41 are part of the hydrophobic pocket involved in interaction with CapRel^SJ46 (Extended Data Figs. 8f and 5b) in the crystal structure. D6 is disordered in the complex, so the D6G substitution may either disrupt an interaction not captured in the crystal structure or stabilize the unbound state of Gp54^Bas11 to prevent CapRel^SJ46 activation.

Finally, when Gp54^Bas11 was introduced to wild-type SECΦ27 such that both wild-type MCP and Gp54^Bas11 were present in the SECΦ27 genome, it led to a stronger defence phenotype (over 10^6-fold reduction in EOP) compared with phages encoding a single activator, and no spontaneous escape mutants were observed (Fig. 5c and Extended Data Fig. 8c). These results indicated that Gp54^Bas11 had functioned as a potent activator when introduced into a related phage SECΦ27, and harbouring both activators (Gp54^Bas11 and the MCP) in its genome rendered this phage extremely sensitive to CapRel^SJ46 defence, as with the native Bas11 phage. By sensing two activators encoded in a single phage genome, CapRel^SJ46 can provide strong defence and limit the ability of phages to escape defence.

## Discussion

Previously, we demonstrated that CapRel^SJ46, a fused toxin–antitoxin system, provides *E. coli* with robust antiphage defence by sensing the MCP of many phages[11] (Fig. 5f). Here we discovered an additional protein trigger for CapRel^SJ46 in the phage Bas11. This alternative trigger,

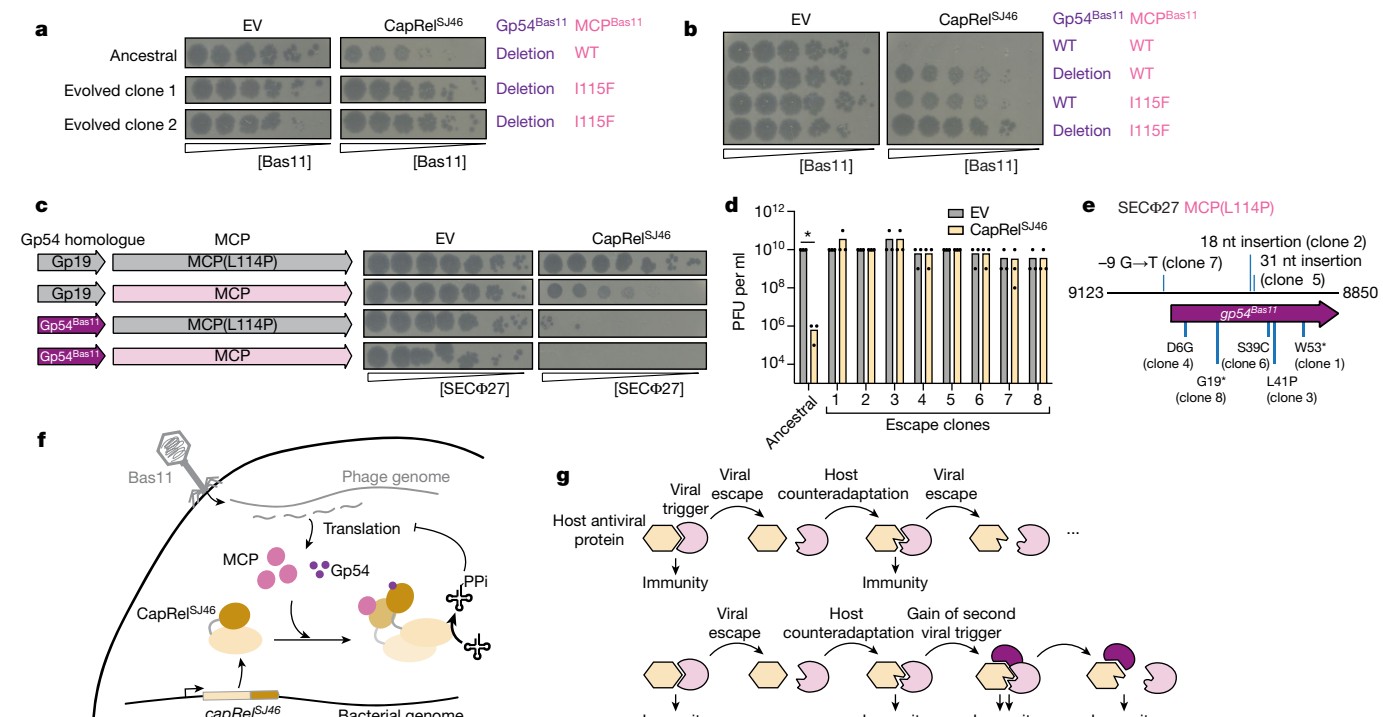

**Fig. 5 | CapRel^SJ46 can sense and respond to two unrelated trigger proteins in Bas11 phage. a**, Serial dilutions of the ancestral and evolved Bas11 phage spotted on lawns of cells harbouring either an EV or a plasmid producing CapRel^SJ46. The corresponding genotypes of its MCP and Gp54 are indicated on the right. **b**, Serial dilutions of the indicated Bas11 phage spotted on lawns of cells harbouring either an EV or a plasmid producing CapRel^SJ46. The corresponding genotypes of its MCP and Gp54 are indicated on the right. **c**, Left, schematics of the region encoding either Gp19 (the Gp54 homologue from SECΦ27) or the MCP. Right, serial dilutions of the indicated SECΦ27 phage spotted on lawns of cells harbouring either an EV or a plasmid producing CapRel^SJ46. **d**, Summary of plaque-forming units (PFU) obtained for the ancestral SECΦ27 MCP(L114P) phage carrying Gp54^Bas11 or eight escape clones, following spotting onto cells

producing CapRel^SJ46 or harbouring an EV. Three independent replicates are shown. *$P = 10^{-18}$ (unpaired two-tailed $t$-test). **e**, Schematic of the gene *19* genomic region in SECΦ27 MCP(L114P) phage replaced by gene *54* from Bas11, with mutations in the escape clones from **d** labelled. **f**, Model for CapRel^SJ46 activation. During infection, newly synthesized MCP and Gp54 bind the antitoxin domain of CapRel^SJ46 to stabilize an active conformation; CapRel^SJ46 then pyrophosphorylates (PPi) tRNAs to inhibit translation and restrict viral replication. **g**, Top, schematic of a conventional Red Queen dynamic between an antiviral immunity protein such as CapRel^SJ46 and a single viral trigger protein, following the schematic in ref. 17. Bottom, when two proteins are recognized by an antiviral system like CapRel^SJ46, viral escape is more difficult because any single escape mutation will not prevent activation of immunity.

Gp54^Bas11, is a small protein of 66 amino acids with unknown function. Despite lacking sequence and structural similarity to the MCPs, Gp54^Bas11 binds to the C-terminal antitoxin domain of CapRel^SJ46 and directly activates it, like the MCPs (Fig. 5f). The interfaces used by the two protein activators overlap but also involve distinct regions. These findings highlight the versatility of a zinc-finger-like domain in interacting with multiple, structurally different proteins. Using such a promiscuous, yet still selective, domain as a phage-infection sensor enables a single defence protein to respond to more than one phage trigger.

Unlike the MCP, which is a conserved and essential structural element of the phage, Gp54^Bas11 is a small protein of unknown function that is not essential to phage Bas11 under laboratory conditions. Gp54^Bas11 might benefit the phage in the wild, possibly by inhibition of another defence system. Recent work has identified other small, non-essential phage proteins that activate one antiphage defence system while also inhibiting another. For example, the Ocr protein of phage T7 inhibits restriction–modification systems, but can also activate the PARIS defence system[26–28].

Although it is often assumed that bacterial defence systems and their triggers will have one-to-one relationships, like most PRRs and PAMPs in eukaryotes, our work demonstrates that CapRel^SJ46 can directly and simultaneously—that is, during a single infection—detect two different proteins (MCP and Gp54^Bas11) from phage Bas11 (Fig. 5f). The detection of multiple phage factors is probably a common feature of bacterial immunity proteins. For instance, a high-throughput screen for

bacteria-encoded triggers of an antiphage retron system identified multiple genes from prophages[19]. Each was sufficient, when overproduced, to trigger the retron, but whether they each contributed to activation during infection is unknown. Similarly, multiple phage proteins other than Ocr can stimulate the PARIS defence system following overexpression[28]. Phages can escape defence with mutations in genes encoding these proteins, but whether they bind directly and whether the binding is similar to or different from Ocr is not yet known[28]. Finally, individual phage mutants often provide incomplete escape from a given defence system[7]; although this may reflect the inability of single mutations to evade defence while still maintaining the phage gene's function, it may also indicate that a second trigger exists. Some defence proteins may even detect more than two phage proteins. Indeed, CapRel^SJ46 protects against T-even phages that lack a homologue of Gp54^Bas11 and produce capsid proteins very different to those in Bas11 and SECΦ27[11], indicating the existence of yet other triggers.

Detection of multiple triggers could render cells susceptible to 'autoimmunity' if a promiscuous defence protein is inadvertently triggered by any related, host-encoded proteins in the absence of an infection. Notably though, at least for CapRel^SJ46, the phage proteins being sensed do not share close relatives in bacterial genomes. Detection of phage proteins without closely related host proteins may be a key feature of bacterial immunity, but how defence proteins balance the sensitivity of detection with the risk of autoimmunity remains to be studied. Sensing of multiple proteins may offer at least three advantages to the immune system: (1) for phages that produce multiple

triggers, the immunity system can provide stronger defence than if it had sensed only one protein; (2) dual sensing makes it more difficult for the phage to completely evade defence unless both triggers are mutated; and (3) sensing of multiple phage proteins may enable protection against a broader set of phages, some of which encode only one trigger or the other. Given these advantages, we anticipate that many antiphage defence systems have evolved similar versatility and also directly sense multiple proteins. Some eukaryotic restriction factors may also sense multiple, unrelated proteins. For instance, TRIM5α, which directly binds retroviral capsid proteins, may also bind the capsid proteins of some DNA viruses[29–31]; and MxA may bind diverse, structurally dissimilar nucleoproteins from RNA viruses[32,33]. The sensing of multiple viral proteins in bacteria or eukaryotes probably leads to complex coevolutionary dynamics between hosts and their viral predators (Fig. 5g). The Red Queen dynamic underpinning host–pathogen relationships is often portrayed as the successive coevolution of two interacting proteins[17], but may involve multiple proteins stemming from many-to-one relationships between triggers and immunity proteins.

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

## Methods

### Strains and growth conditions

All bacterial and phage strains used in this study are listed in Supplementary Table 1. *E. coli* strains were routinely grown at 37 °C in Luria broth (LB) medium for cloning and maintenance. Phages were propagated by infecting a culture of *E. coli* MG1655 at an optical density ($OD_{600}$) of around 0.1–0.2 and multiplicity of infection of 0.1. Cleared cultures were pelleted by centrifugation to remove residual bacteria and filtered through a 0.2 μm filter. Chloroform was then added to phage lysates for prevention of bacterial growth. All phage-infection experiments were performed in LB medium at 25 °C. Antibiotics were used at the following concentrations (liquid, plates): carbenicillin (50, 100 μg ml⁻¹); chloramphenicol (20, 30 μg ml⁻¹).

### Plasmid construction

All plasmids are listed in Supplementary Table 2, and all primers in Supplementary Table 3.

**pBAD33-gp54^Bas11 constructs.** Wild-type or mutant variants of gp54^Bas11 were PCR amplified from the corresponding wild-type Bas11 or escaping phage clones using primers TZ-3 and TZ-4, and inserted into pBAD33 linearized with TZ-1 and TZ-2 using Gibson assembly. To add a C-terminal HA-tag, primers TZ-9 and TZ-10 were used to PCR amplify pBAD33-gp54^Bas11 followed by Gibson assembly. Mutations that produce the single amino acid substitutions D6E, A7V and I25V in Gp54^Bas11 were generated by site-directed mutagenesis using primers TZ-38–43.

**pBR322-capRel^SJ46 constructs.** Mutations that produce the single amino acid substitutions N275D, D273K, K278E and S279P were generated by site-directed mutagenesis using primers TZ-11–18. Mutations that produce the triple substitution K278E/R314E/K316E were introduced by two-step, site-directed mutagenesis using primers TZ-23–24, then TZ-15 and TZ-16. To add a C-terminal FLAG-tag to CapRel^SJ46, primers TZ-25 and TZ-26 were used to PCR amplify pBR322-capRel^SJ46 followed by Gibson assembly.

**pBAD33-gp54^Bas11 homologue constructs.** The genes encoding Gp54^Bas11 homologues (Gp57^Bas10, Gp60^Bas8, Gp57^Bas5 and Gp19^SECΦ27) were PCR amplified from the corresponding phage using primers TZ-30–37 and inserted into linearized pBAD33 by Gibson assembly. Mutations that produce the single amino acid substitutions V7A and V25I in Gp57^Bas10 were generated by site-directed mutagenesis using primers TZ-44–47.

**pET-His₆-gp54^Bas11 constructs.** Either wild-type or the G24D variant of gp54^Bas11 was PCR amplified from the corresponding phage using primers TZ-7 and TZ-8, and inserted into pET-His₆ vector linearized with primers TZ-5 and TZ-6 using Gibson assembly.

**pET-His₆-MBP-capRel^SJ46 construct.** capRel^SJ46 was first PCR amplified from pBR322-capRel^SJ46 using primers TZ-48 and TZ-49, and inserted into pET-His₆ vector linearized with primers TZ-5 and TZ-6 using Gibson assembly. The gene encoding MBP was PCR amplified with TZ-52 and TZ-53, and inserted into pET-His₆-capRel^SJ46 linearized with primers TZ-50 and TZ-51 using Gibson assembly.

**pBAD33-mcp^Bas10 construct.** The gene encoding the MCP of Bas10 (Gp9^Bas10) was PCR amplified from phage Bas10 using primers TZ-27 and TZ-28, and inserted into pBAD33 linearized with primers TZ-29 and TZ-1 using Gibson assembly.

**pBR322-gp54^Bas11-Flag constructs.** gp54^Bas11 with its native promoter was PCR amplified from phage Bas11 using primers TZ-58 and TZ-59, and inserted into pBR322-capRel^SJ46-Flag linearized with primers TZ-60

and TZ-61 using Gibson assembly. Corresponding upstream mutations were introduced using site-directed mutagenesis.

### Strain construction

Plasmids described above were introduced into *E. coli* MG1655 by TSS transformation or electroporation.

Bas11 mutant phage producing MCP(I115F) was generated using a CRISPR–Cas system for targeted mutagenesis as described previously[34]. In brief, sequences for RNA guides used to target Cas9-mediated cleavage were designed using the toolbox in Geneious Prime 2022.0.2 and selected for targeting of mcp^Bas11 (Gp8 in Bas11), but nowhere else in the Bas11 genome. Guides were inserted into the pCas9 plasmid and tested for their ability to restrict Bas11. An efficient guide was selected and the pCas9-guide plasmid was cotransformed into *E. coli* MG1655 with a high-copy-number repair plasmid containing mcp^Bas11(I115F), with the guide mutated synonymously to prevent self-cutting. The wild-type Bas11 phage was plated onto a strain containing both the pCas9-guide and the repair plasmid, and single plaques were screened by Sanger sequencing. Two clones that produce the I115F-substituted MCP^Bas11 were propagated on strains containing only pCas9-guide for further selection.

SECΦ27 mutant phages producing Gp54^Bas11 rather than its homologue in SECΦ27 (Gp19) were generated as described above. The guide was selected such that it targeted only gene *19* in SECΦ27, but not gene *54* from Bas11. The selected pCas9-guide plasmid was cotransformed into *E. coli* MG1655 with a high-copy-number repair plasmid containing the coding sequence of gene *54* from Bas11, flanked by the region that flanks gene *19* in SECΦ27 for homologous recombination. Either the wild-type SECΦ27 phage or the mutant producing MCP(L114P) was plated onto the strain containing pCas9 plasmid and the repair plasmid for selection. Two clones each were propagated and selected twice on strains containing only pCas9-guide.

### Phage-spotting assays and EOP measurements

Phage-spotting assays were conducted similarly to a method described previously[11]. Phage stocks isolated from single plaques were propagated in *E. coli* MG1655 at 37 °C in LB. To titre phage, dilutions of stocks were mixed with *E. coli* MG1655 and melted LB + 0.5% agar, spread on LB + 1.2% agar plates and incubated at 37 °C overnight. For phage-spotting assays, 80 μl of a bacterial strain of interest was mixed with 4 ml of LB + 0.5% agar and spread on an LB + 1.2% agar + antibiotic plate. Phage stocks were then serially diluted in 1× FM buffer (20 mM Tris-HCl pH 7.4, 100 mM NaCl, 10 mM MgSO₄), and 2 μl of each dilution was spotted on the bacterial lawn. Plates were then incubated at 25 °C overnight before imaging. EOP was calculated by comparing the ability of the phage to form plaques on an experimental strain relative to the control strain. Experiments were replicated three times independently, and representative images are shown.

### Toxicity assays on solid media

Bacterial toxicity assays were conducted similarly to a method described previously[11]. For coproduction of CapRel^SJ46 with either Gp54 homologues or MCPs, single colonies of *E. coli* MG1655 harbouring pBR322-capRel^SJ46 and pBAD33-gp54 homologue or pBAD33-mcp (wild-type or the corresponding variants) were grown for 6 h at 37 °C in LB-glucose to saturation. Next, 200 μl of each saturated culture was pelleted by centrifugation at 4,000*g* for 10 min, washed once in 1× PBS and resuspended in 400 μl of 1× PBS. Cultures were then serially diluted tenfold in 1× PBS and spotted on M9L plates supplemented with 0.4% glucose or 0.2% arabinose. M9L plates contain M9 medium (6.4 g l⁻¹ Na₂HPO₄-7H₂O, 1.5 g l⁻¹ KH₂PO₄, 0.25 g l⁻¹ NaCl, 0.5 g l⁻¹ NH₄Cl medium supplemented with 0.1% casamino acids, 0.4% glycerol, 2 mM MgSO₄ and 0.1 mM CaCl₂) supplemented with 5% LB (v/v). Plates were then incubated at 37 °C overnight before imaging.

## Isolation of phage escape mutants to infect CapRel[SJ46]

Bas11 or SECΦ27 MCP(L114P) escape mutants were isolated by plating a population of phage onto CapRel[SJ46]-containing cells. Next, 20 µl of $10^{10}$ PFU ml$^{-1}$ Bas11 or SECΦ27 MCP(L114P) phage, mixed with 40 µl of overnight culture of *E. coli* MG1655 pBR322-capRel[SJ46], was added to 4 ml of LB + 0.5% agar and spread onto LB + 1.2% agar. Plates were incubated at 25 °C overnight. Single plaques were isolated and propagated using the same strain in LB at 25 °C. Amplified phage lysates were pelleted to remove bacteria, and sequenced by Illumina sequencing as described below to identify mutations.

Bas10 or Bas11 phage with gene *54* deleted was evolved to completely overcome CapRel[SJ46] defence using an experimental evolution protocol described previously[35]. In brief, five independent populations were evolved in a 96-well plate containing a sensitive host *E. coli* MG1655 pBR322-EV and a resistant host *E. coli* MG1655 pBR322-capRel[SJ46]. One control population was evolved with only the sensitive host. Overnight bacterial cultures were back-diluted to an $OD_{600}$ of 0.01 in LB, and 100 µl was seeded into each well. Cells were infected with tenfold serial dilutions of Bas10 or Bas11 phage with gene *54* deleted, at multiplicity of infection $100-10^{-4}$, with one well uninfected to monitor for contamination. Plates were sealed with breatheable plate seals and incubated at 25 °C for either 14 h (for Bas10) or 17 h (for Bas11) in a plate shaker at 1,000 rpm. Cleared wells from each population were pooled, pelleted at 4,000*g* for 20 min to remove bacteria and supernatant lysates were transferred to a 96-deep-well block with 40 µl of chloroform added to prevent bacterial growth. Lysates were spotted onto both sensitive and resistant hosts to check the defence phenotype. Three rounds of evolution were performed for Bas10, and four populations were able to overcome CapRel[SJ46] defence. Two rounds of evolution were performed for Bas11 phage with gene *54* deleted. Evolved clones from each evolved population were isolated by plating to single plaques on lawns of resistant host, and control clones from the control population were isolated on a lawn of the sensitive host. Two clones from each population were propagated using the corresponding host and sequenced as described below.

## Phage DNA extraction and Illumina sequencing

Phage DNA extraction and sequencing were conducted as described previously[11]. To extract phage DNA, high-titre phage lysates (over $10^6$ PFU µl$^{-1}$) were treated with DNase I (0.001 U µl$^{-1}$) and RNase A (0.05 mg ml$^{-1}$) at 37 °C for 30 min, then 10 mM EDTA was used to inactivate the nucleases. Lysates were then incubated with Proteinase K at 50 °C for 30 min to disrupt capsids and release phage DNA, which was isolated by ethanol precipitation. In brief, sodium acetate pH 5.2 was added to 300 mM followed by 100% ethanol to yield a final volume fraction of 70%. Samples were incubated at −80 °C overnight, pelleted at 21,000*g* for 20 min and supernatant removed. Pellets were washed with 100 µl of isopropanol and 200 µl of 70% (v/v) ethanol, then air-dried at room temperature and resuspended in 25 µl of 1× TE buffer (10 mM Tris-HCl, 0.1 mM EDTA, pH 8.0). Concentrations of extracted DNA were measured by NanoDrop (Thermo Fisher Scientific).

For preparation of Illumina sequencing libraries, 100–200 ng of genomic DNA was sheared in a Diagenode Bioruptor 300 sonicator water bath for 20 × 30 s cycles at maximum intensity. Sheared gDNA was purified using AMPure XP beads, followed by end repair, 3′ adenylation and adaptor ligation. Barcodes were added to both 5′ and 3′ ends by PCR with primers that anneal to the Illumina adaptors. The libraries were cleaned by AMPure XP beads using a double cut to elute fragment sizes matching the read lengths of the sequencing run. Libraries were sequenced on an Illumina MiSeq at the MIT BioMicro Center. Illumina reads were assembled to the reference genomes using Geneious Prime 2022.0.2.

## Coimmunoprecipitation analysis

Coimmunoprecipitation experiments were conducted similarly to those described previously[11]. For coproduction of CapRel[SJ46] and Gp54[Bas11] or with MCP[SECΦ27], *E. coli* MG1655 containing pBR322-capRel[SJ46] or pBR322-capRel[SJ46]-Flag (wild-type or mutant variants) and pBAD33-gp54[Bas11]-HA (wild-type or mutant variants) or pBAD33-mcp[SECΦ27]-HA were grown overnight in M9-glucose. Overnight cultures were back-diluted to an $OD_{600}$ of 0.05 in 50 ml of M9 (no glucose) and grown to an approximate $OD_{600}$ of 0.3 at 37 °C. Cells were induced with 0.2% arabinose for 30 min at 37 °C, then $OD_{600}$ was measured and cells pelleted at 4,000*g* for 10 min at 4 °C. Supernatant was removed and cells resuspended in 800 µl of lysis buffer (25 mM Tris-HCl, 150 mM NaCl, 1 mM EDTA, 1% Triton X-100, 5% glycerol) supplemented with protease inhibitor (Roche), 1 µl ml$^{-1}$ Ready-Lyse Lysozyme Solution (Lucigen) and 1 µl ml$^{-1}$ benzonase nuclease (Sigma). Samples were lysed by two freeze–thaw cycles, and lysates normalized by $OD_{600}$. Lysates were pelleted at 21,000*g* for 10 min at 4 °C, and 750 µl of supernatant was incubated with prewashed anti-Flag magnetic agarose beads (Pierce) for 1 h at 4 °C with end-over-end rotation. Beads were then washed three times with 500 µl of lysis buffer, followed by the direct addition of 1× Laemmli sample buffer (Bio-Rad) supplemented with 2-mercaptoethanol to beads to elute proteins. Samples were boiled at 95 °C, analysed by 4–20% SDS–PAGE and transferred to a 0.2 µm polyvinylidene difluoride membrane. Anti-Flag and anti-HA antibodies (Cell Signaling Technology) were used at a final concentration of 1:1,000, and SuperSignal West Femto Maximum Sensitivity Substrate (ThermoFisher) was used to develop blots. Blots were imaged by the ChemiDoc Imaging system (Bio-Rad). Images shown are representatives of two independent biological replicates.

## Immunoblot of Gp54[Bas11] expression levels

Single colonies of *E. coli* MG1655 pBR322-gp54[Bas11]-Flag with its wild-type or mutant native promoter were grown overnight in LB. Overnight cultures were back-diluted to $OD_{600}$ of 0.05 in 10 ml of fresh LB and grown to $OD_{600}$ of 0.4 at 37 °C. $OD_{600}$ was measured, and 5 ml of cells pelleted at 4,000*g* for 5 min with $OD_{600}$ normalized. Supernatant was removed and pellets resuspended in 1× Laemmli sample buffer (Bio-Rad) supplemented with 2-mercaptoethanol. Samples were then boiled at 95 °C and analysed using 4–20% SDS–PAGE and transferred to a 0.2 µm polyvinylidene difluoride membrane. Anti-Flag antibody (Cell Signaling Technology) and anti-RpoA antibody (BioLegend) were used at a final concentration of 1:1,000, and SuperSignal West Femto Maximum Sensitivity Substrate (ThermoFisher) was used to develop blots. Blots were imaged using a ChemiDoc Imaging system (Bio-Rad). Images shown are representative of two independent biological replicates.

## Error-prone PCR mutagenesis of CapRel[SJ46] and selection with Gp54[Bas11]

The C terminus of CapRel[SJ46] was mutagenized using error-prone PCR-based mutagenesis as described previously[11]. In brief, primers TZ-54 and TZ-55 were used to amplify the C terminus of CapRel[SJ46] using Taq polymerase (NEB), with 0.5 mM MnCl$_2$ added to the reaction as the mutagenic agent. PCR products were treated with Dpn I, column purified and inserted into a pBR322-capRel[SJ46] backbone amplified with primers TZ-56 and TZ-57 using Gibson assembly. Gibson products were transformed into DH5α and grown overnight in LB at 37 °C. Overnight cultures were miniprepped to obtain the mutagenized library, and individual colonies were Sanger sequenced to assess the number of mutations. To perform selection, the mutagenized library was electroporated into *E. coli* MG1655 pBAD33-gp54[Bas11] and plated onto M9L plates containing 0.2% arabinose to select for survivors. Colonies were picked and sequenced to identify mutations in CapRel[SJ46].

## Protein expression and purification

For the production of His$_6$-MBP-tagged CapRel[SJ46], *E. coli* BL21(DE3) cells were transformed with pET-His$_6$-MBP-capRel[SJ46] and grown in LB medium to $OD_{600}$ = 0.5. Protein expression was induced by the addition of 0.3 mM isopropyl-β-D-thiogalactopyranoside, and cells grown for 3 h at 30 °C. The culture was centrifuged at 4,000*g* for 10 min at 4 °C

and the cell pellet resuspended in lysis buffer (50 mM Tris-HCl pH 8.0, 500 mM NaCl, 500 mM KCl, 2 mM MgCl$_2$, 1 mM DTT) supplemented with 0.4 mM phenylmethanesulfonyl fluoride, 10 µg ml$^{-1}$ lysozyme and 7.5 U ml$^{-1}$ benzonase nuclease (Millipore). Cells were disrupted using sonication (Qsonica), and glycerol was added to the lysate at a final 10% concentration following sonication. The supernatant was separated from the pellet by centrifugation (15,000 rpm for 30 min, JA-25.50 rotor, Beckman Coulter). The clarified supernatant was loaded onto a gravity-flow chromatography column (Bio-Rad) packed with 2 ml of Ni-NTA agarose resin (Qiagen) pre-equilibrated with 15 ml of lysis buffer. The resin was washed with ten column volumes of wash buffer 1 (50 mM Tris-HCl pH 8.0, 500 mM NaCl, 500 mM KCl, 2 mM MgCl$_2$, 10 mM imidazole, 10% glycerol, 1 mM DTT) and then with ten column volumes of wash buffer 2 (50 mM Tris-HCl pH 8.0, 500 mM NaCl, 500 mM KCl, 2 mM MgCl$_2$, 50 mM imidazole, 10% glycerol, 1 mM DTT). Proteins were eluted in 4 ml of elution buffer (50 mM Tris-HCl pH 8.0, 500 mM NaCl, 500 mM KCl, 2 mM MgCl$_2$, 300 mM imidazole, 10% glycerol, 1 mM DTT). For removal of any remaining contaminants, the eluted protein sample was loaded onto a size exclusion chromatography (SEC) Superdex 200 Increase 10/300 GL column (Cytiva) pre-equilibrated in SEC buffer (50 mM Tris-HCl pH 8.0, 250 mM NaCl, 250 mM KCl, 2 mM MgCl$_2$, 1 mM DTT). Fractions containing the protein of interest were pooled and concentrated to around 1 mg ml$^{-1}$. Purity of protein samples was assessed both spectrophotometrically and by SDS–PAGE.

To produce His$_6$-tagged Gp54$^{Bas11}$ or the G24D variant, *E. coli* BL21(DE3) cells were transformed with pET-His$_6$-gp54$^{Bas11}$ (wild-type or G24D) and grown in LB medium to an OD$_{600}$ of 0.5. Protein expression was induced by the addition of 0.3 mM isopropyl-β-D-thiogalactopyranoside, and cells were grown for 3 h at 30 °C. Purification steps were performed similarly to those described above, with the following buffers. Lysis buffer contained 50 mM Tris-HCl pH 8.0, 150 mM NaCl, 2 mM MgCl$_2$ and 1 mM DTT supplemented with 0.4 mM phenylmethanesulfonyl fluoride, 10 µg ml$^{-1}$ lysozyme and 7.5 U ml$^{-1}$ benzonase nuclease (Millipore). Wash buffer 1 contained 50 mM Tris-HCl pH 8.0, 500 mM NaCl, 2 mM MgCl$_2$, 10 mM imidazole, 10% glycerol and 1 mM DTT. Wash buffer 2 contained 50 mM Tris-HCl pH 8.0, 150 mM NaCl, 2 mM MgCl$_2$, 50 mM imidazole, 10% glycerol and 1 mM DTT. Elution buffer contained 50 mM Tris-HCl pH 8.0, 150 mM NaCl, 2 mM MgCl$_2$, 300 mM imidazole, 10% glycerol and 1 mM DTT. To remove any remaining contaminants, the eluted protein sample was loaded onto a SEC Superose 6 Increase 10/300 GL column (Cytiva) pre-equilibrated in SEC buffer (50 mM Tris-HCl pH 8.0, 150 mM NaCl, 2 mM MgCl$_2$, 1 mM DTT). Fractions containing the protein of interest were pooled and concentrated to around 5 mg ml$^{-1}$. Purity of protein samples were assessed both spectrophotometrically and by SDS–PAGE.

### Cell-free translation

Experiments using the PURExpress in vitro protein synthesis kit (NEB, E6800) were performed as per the manufacturer's instructions. All reactions were supplemented with 0.8 U µl$^{-1}$ RNase Inhibitor Murine (NEB, M0314S). Purified His$_6$-MBP-tagged CapRel$^{SJ46}$ protein was added to the reaction at a final concentration of 500 nM, and either purified His$_6$-tagged Gp54$^{Bas11}$ or the G24D variant was used at a final concentration of 4 µM. A template plasmid encoding the control protein DHFR (provided by the kit) was used at 6 ng µl$^{-1}$. The reactions were incubated at 37 °C for 2 h, and 2 µl of each reaction was mixed with 10 µl of 1× Laemmli sample buffer (Bio-Rad) supplemented with 2-mercaptoethanol. Mixtures were boiled for 5 min at 95 °C and analysed by 12% SDS–PAGE. Gels were stained with Coomassie stain and imaged using the ChemiDoc Imaging system (Bio-Rad). Images shown are representative of three independent biological replicates.

### Crystallization and structure determination of Gp54$^{Bas11}$ and the CapRel$^{SJ46}$–Gp54$^{Bas11}$ complex

His$_6$-tagged Gp54$^{Bas11}$ was purified as described above and concentrated to 5 mg ml$^{-1}$ for crystallization. Crystallization conditions for

His$_6$–Gp54$^{Bas11}$ were screened by sitting-drop vapour diffusion using a Formulatrix NT8 drop setter and commercial screening kits. Each drop, consisting of 100 nl of protein solution plus 100 nl of reservoir solution, was equilibrated against 70 µl of reservoir solution. Crystals appeared in Index HT (Hampton Research) condition B12 (2.8 M sodium acetate trihydrate pH 7.0). These conditions were optimized, and the final crystals were grown by hanging-drop vapour diffusion, with drops consisting of 2 µl of protein plus 2 µl of well solution (3.2 M potassium acetate pH 7.0) at room temperature. After 8 days, a crystal was harvested and directly vitrified in a nitrogen gas stream at 100 K (Oxford Cryostream). X-ray diffraction data were collected on a Rigaku Micromax-007 rotating anode with Osmic VariMax-HF mirrors and a Rigaku Saturn 944 detector. Diffraction data were processed with the XDS suite[36]. Phaser[37] was used to solve the structure by molecular replacement using an AlphaFold[38] model. The molecular replacement solution was refined in PHENIX[39] with manual model building done with Coot[40]. The model was refined to a final $R_{work}/R_{free}$ of 0.211/0.252. X-ray data collection and refinement statistics are summarized in Extended Data Table 1.

For the CapRel$^{SJ46}$–Gp54$^{Bas11}$ complex, CapRel$^{SJ46}$ and Gp54$^{Bas11}$ were purified as described above and mixed in a 1:1 ratio at a concentration of 2 mg ml$^{-1}$. The complex was then further purified by SEC (in 50 mM Tris-HCl pH 8.0, 150 mM NaCl, 2 mM MgCl$_2$ and 1 mM DTT) and the resulting sample concentrated to 10 mg ml$^{-1}$ for crystallization. Crystallization conditions for the CapRel$^{SJ46}$–Gp54$^{Bas11}$ complex were either screened as such or supplemented with 5 mM ATP. Crystals grew within 1 week in 25% PEG 1500 in a malic acid, MES, Tris buffer system (pH 8.0). Before data collection, crystals were cryoprotected by soaking in the mother liquor solution supplemented with 25% glycerol and flash-frozen in liquid nitrogen for storage. X-ray diffraction data were collected at the I24 beamline of the Diamond Light Source synchrotron (UK) on a CdTe Eiger2 9M detector, then processed using the XDS suite[36] and scaled with Aimless. The structure was solved by molecular replacement performed with Phaser[37] using the coordinates of the toxSYNTH domain of CapRel$^{SJ46}$ (PDB: 7ZTB). Initial automated model building was performed with Buccaneer[41], which partially completed Gp54$^{Bas11}$ and further improved with the MR-Rosetta suite from the Phenix package[42]. Following several iterations of manual building with Coot[40] and maximum-likelihood refinement as implemented in Buster/TNT[43], the model was refined to $R_{work}/R_{free}$ of 0.193/0.236. X-ray data collection and refinement statistics are summarized in Extended Data Table 1.

### Homology search, alignment and conservation analysis

CapRel$^{SJ46}$ homologues were identified, aligned and used as input for ConSurf analysis as described previously[11]. Homologues of the MCPs or Gp54$^{Bas11}$ in BASEL phages were identified by BLASTp[44] searches against each phage genome, and aligned by MUSCLE[45]. Whole genomes of phages were aligned using Mauve[46] in Geneious Prime 2022.0.2.

### Structure prediction

The structure prediction of CapRel$^{SJ46}$-MCP$^{SECΦ27}$ was reported previously[11] and was calculated using AlphaFold2. The structure of CapRel$^{SJ46}$ in the closed state for comparison with the experimental SAXS curve was also calculated using AlphaFold2 using default parameters (as implemented in ColabFold[38]) and running the calculations for ten recycles. Both models are deposited in the ModelArchive Database (https://www.modelarchive.org) with the accession codes ma-zblch (https://doi.org/10.5452/ma-zblch) and ma-9z23e (https://doi.org/10.5452/ma-9z23e).

### Circular dichroism spectroscopy

Circular dichroism measurements were performed on a MOS-500 spectropolarimeter (BioLogic) using a cuvette of 0.1 cm path length. Spectra were collected between 200 and 250 nm with a data interval of 0.25 nm at 25 °C. Measurements were recorded in 15 mM

$K_2HPO_4$, 0.05 mM $KH_2PO_4$ pH 7.5, 300 mM KCl, 300 mM NaCl and 1 mM tris(2-carboxyethyl)phosphine (TCEP). Protein concentration used in measurements was 0.6 mg ml$^{-1}$. Molar residue ellipticities ($\theta$, mdeg cm$^2$ dmol$^{-1}$) were obtained from the raw data ($\theta$, ellipticity) following buffer correction, according to the relation $[\theta] = \theta M_w/(ncl)$, where $M_w$ is weight-averaged molecular mass, $c$ mass concentration, $l$ optical path length and $n$ the number of amino acid residues.

### Hydrogen deuterium exchange mass spectrometry

Hydrogen deuterium exchange mass spectrometry (HDX–MS) experiments were performed on an HDX platform comprising a Synapt G2 mass spectrometer (Waters Corporation) connected to a nanoAcquity ultra-performance liquid chromatography (UPLC) system following the protocol previously described[11]. Samples of CapRel$^{SJ46}$, Gp54$^{Bas11}$ and CapRel$^{SJ46}$–Gp54$^{Bas11}$ were prepared at a concentration of 100 µM (the integrity of the complex was confirmed by SEC before the HDX–MS experiment). For each experiment, 8 µl of sample was incubated for 1, 5, 15 and 60 min in 72 µl of labelling buffer L (50 mM HEPES, 500 mM KCl, 500 mM NaCl, 2 mM $MgCl_2$, 1 mM TCEP, 0.002% mellitic acid, pD 7.5) at 20 °C. Non-deuterated reference points were prepared by replacement of buffer L with equilibration buffer E (50 mM HEPES, 500 mM KCl, 500 mM NaCl, 2 mM $MgCl_2$, 1 mM TCEP, 0.002% mellitic acid, pH 7.5). After labelling, samples were quenched by mixing with 80 µl of prechilled quench buffer Q (50 mM $K_2PO_4$, 1 mM TCEP, 1.2% formic acid, pH 2.4). Samples were then directly flash-frozen in liquid nitrogen and stocked at −80 °C until injection. For injection, samples were thawed at room temperature and 150 µl of quench samples directly transferred to a Enzymate BEH Pepsin Column (Waters Corporation) at 200 µl min$^{-1}$ and 20 °C, with a pressure 3,000 pounds per square inch. Peptic peptides were trapped for 3 min on an Acquity UPLC BEH C18 VanGuard Pre-column (Waters Corporation) at a flow rate of 200 µl min$^{-1}$ in water (0.1% formic acid in high-performance liquid chromatography water, pH 2.5) before elution on an Acquity UPLC BEH C18 Column for chromatographic separation. Separation was performed with a linear gradient buffer (3–45% gradient of 0.1% formic acid in acetonitrile) at a flow rate of 40 µl min$^{-1}$. Peptide identification and deuteration uptake analysis were performed on a Synapt G2, using positive electrospray ionization, data independent acquisition, and triwave ion-mobility for improved resolution and identification. Leucine enkephalin was applied for mass accuracy correction, and sodium formate was used as calibration for the mass spectrometer. MSE data were collected with a 20–30 V transfer collision energy ramp. The pepsin column was washed between injections using pepsin wash buffer (1.5 M guanidinium HCl, 4% (v/v) acetonitrile, 0.8% (v/v) formic acid). A cleaning run was performed on every third sample to prevent peptide carryover. Optimized peptide identification and peptide coverage for all samples were performed from undeuterated controls (five replicates). All deuterium time points were performed in triplicate. The mass spectrometry proteomics data have been deposited to the ProteomeXchange Consortium by the PRIDE[47] partner repository with the dataset identifier PXD050526.

### Small angle X-ray scattering

Samples for small angle X-ray scattering (SAXS) were concentrated to 10 mg ml$^{-1}$, flash-frozen and stored at −80 °C. SAXS data were collected at the SWING beamline (Soleil and ESRF synchrotrons, France) on a Pilatus 2M detector using the standard beamline set-up in SEC mode. Samples were prepared in 500 mM NaCl, 500 mM KCl, 2 mM TCEP and 30 mM HEPES pH 7.5. SEC–SAXS was performed with a Shodex KW404–4 F column coupled to a high-performance liquid chromatography system, in front of the SAXS data collection capillary. Samples were flowed at 0.2 ml min$^{-1}$ and data collected at 10 °C. Radiation-damaged frames were removed before data analysis. Data were analysed with the ATSAS suite[48]. SAXS-based models were derived from the coordinates of the X-ray structure of the CapRel$^{SJ46}$–Gp54$^{Bas11}$ complex and an AlphaFold model of unbound CapRel$^{SJ46}$. Calculation of ab initio shapes was carried out with the program DAMMIF from the ATSAS package.

### Isothermal titration calorimetry

All titrations were performed with an Affinity ITC (TA instruments) at 30 °C. For titration, CapRel$^{SJ46}$ was loaded in the instrument syringe at 200 µM and Gp54$^{Bas11}$ used in the cell at 10 µM. Titrations were performed in 50 mM HEPES pH 7.5, 500 mM KCl, 500 mM NaCl, 2 mM $MgCl_2$ and 1 mM TCEP. Final concentrations were verified by absorption using a Nanodrop One (ThermoScientific). All isothermal titration calorimetry (ITC) measurements were performed by titrating 2 µl of CapRel$^{SJ46}$ into Gp54$^{Bas11}$ (Gp54$^{Bas11}$(G24D) was used at 260 µM) at a constant stirring rate of 75 rpm. All data were processed, buffer corrected and analysed using the NanoAnalyse and Origin software packages.

### Reporting summary

Further information on research design is available in the Nature Portfolio Reporting Summary linked to this article.

### Data availability

Structural data for Gp54$^{Bas11}$ and the CapRel$^{SJ46}$–Gp54$^{Bas11}$ complex are available in the Protein Data Bank (PDB) under accessions 9AXB and 9ERV, respectively. Sequencing data are available in the Sequence Read Archive under BioProject PRJNA1084025. HDX–MS data can be accessed through ProteomeXchange with identifier PXD050526. AlphaFold-predicted structural models are deposited in the ModelArchive Database (https://www.modelarchive.org) with the accession codes ma-zblch (https://doi.org/10.5452/ma-zblch) and ma-9z23e (https://doi.org/10.5452/ma-9z23e). All other data are available in the manuscript or Supplementary Information. Other previously published structures are available in PDB (7ZTB and 5LD2). The UniRef90 database is publicly available. Reference phage genomes are publicly available: SECΦ27 (NC_047938.1), Bas05 (MZ501101.1), Bas08 (MZ501059.1), Bas10 (MZ501077.1), Bas11 (MZ501085.1). Details of materials, including strains and plasmids, are available on reasonable request. Source Data are provided with this paper.

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

**Acknowledgements** We thank A. Harms for sharing the BASEL phage collection, the MIT BioMicro Center and its staff for their support in sequencing and the MIT Structural Biology core and P. Rosen for help with X-ray crystallography. We thank S. Srikant, C. Vassallo, C. Beck, J. Ramseyer and D. Saxton for comments on the manuscript, and all members of the Laub laboratory for helpful discussions. M.T.L. is an Investigator of the Howard Hughes Medical Institute. A.G.-P. was supported by Fonds National de Recherche Scientifique (CDR J.0065.23F, PDR T.0090.22), ERC (CoG DiStRes, 864311), Fonds Jean Brachet, Fondation Van Buuren and FNRS WELBIO ADV X.1520.24F. C.M. was supported by Fonds de la Recherche Scientifique (MIS grant F.45322.22). K.C.W. is a fellow of the FRIA. We acknowledge the use of the SWING beamline at the Soleil synchrotron (Gif-sur-Yvette, France) and I24 (Diamond Light Source synchrotron, UK).

**Author contributions** Experiments were conceived and designed by T.Z., A.G.-P. and M.T.L. All experiments were performed by T.Z. except for X-ray crystallography, which was done together with D.L. and A.G.-P., and ITC and HDX data, which were collected by A.C., A.N., K.C.W. and C.M. SAXS data were collected by A.T. Figure design, manuscript writing and editing were carried out by T.Z., A.G.-P. and M.T.L. Project supervision and funding were undertaken by A.G.-P. and M.T.L.

**Competing interests** A.G.-P. is cofounder and stockholder of Santero Therapeutics. The other authors declare no competing interests.

**Additional information**
**Correspondence and requests for materials** should be addressed to Abel Garcia-Pino or Michael T. Laub.

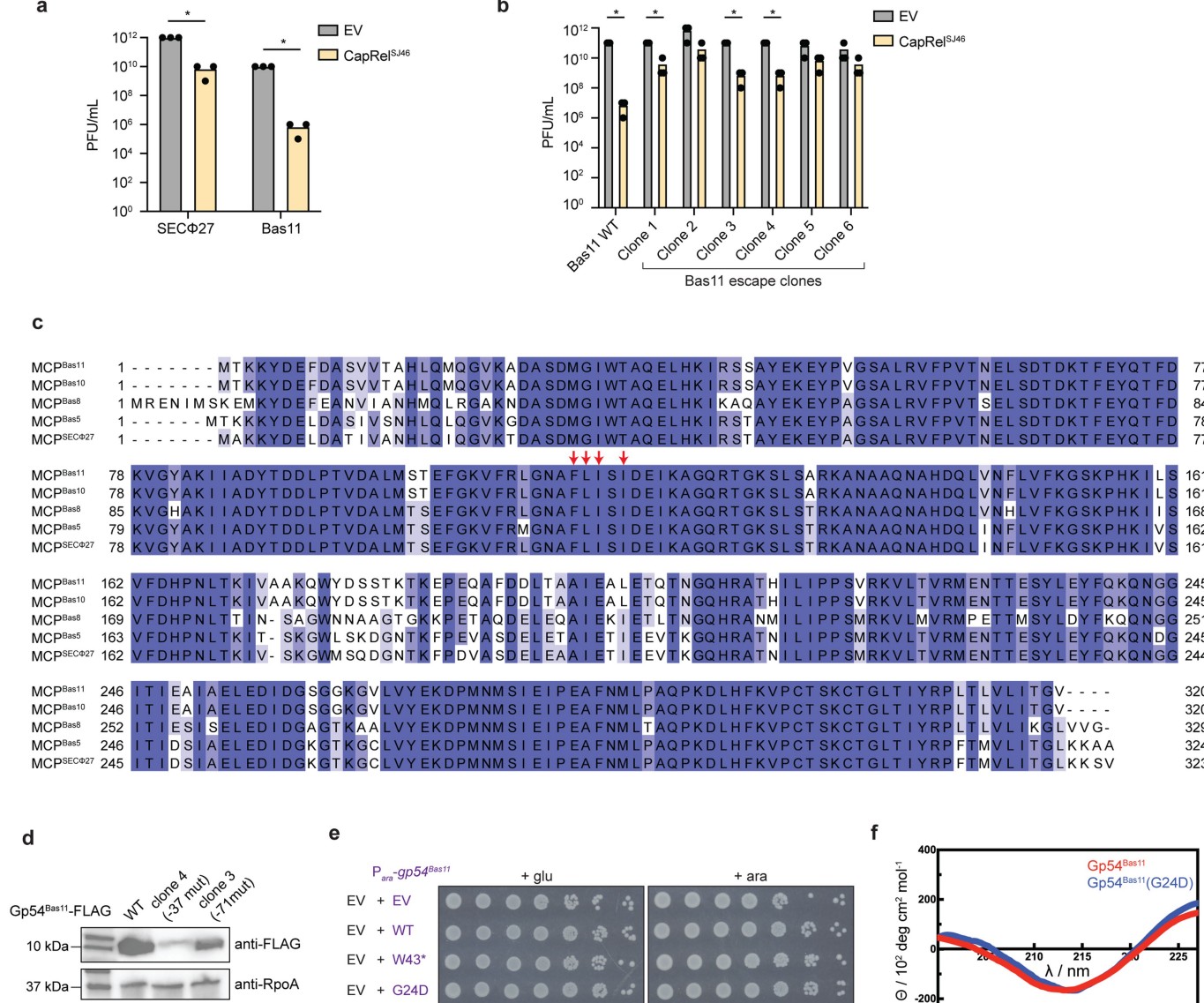

**Extended Data Fig. 1 | Gp54^Bas11 activates CapRel^SJ46.** (**a**) Summary of 3 independent replicates of phage spotting assay in Fig. 1a. Asterisks indicate p = 10^−10 (SECΦ27) or 10^−18 (Bas11) (unpaired two-tailed t-test). (**b**) Summary of 3 independent replicates of phage spotting assay in Fig. 1b. Asterisks indicate p = 10^−18 (WT), 10^−6 (clone 1) or 10^−10 (clone 3 and clone 4) (unpaired two-tailed t-test). (**c**) Multiple sequence alignment of the major capsid proteins from phages SECΦ27, Bas5, Bas8, Bas10 and Bas11. Residues shown to be important for interaction with CapRel^SJ46 are labeled by red arrows. (**d**) Immunoblot of FLAG-tagged Gp54^Bas11 expressed from its native promoter (wild-type or mutated versions found in escape phage clones). RpoA is included as a loading control. Image shown is a representative of 2 biological replicates. (**e**) Cell viability assessed by serial dilutions of cells expressing an empty vector (EV) and the indicated variant of Gp54^Bas11 from an arabinose-inducible promoter on media containing glucose or arabinose. (**f**) Circular dichroism spectra for purified His₆-Gp54^Bas11 and His₆-Gp54^Bas11(G24D).

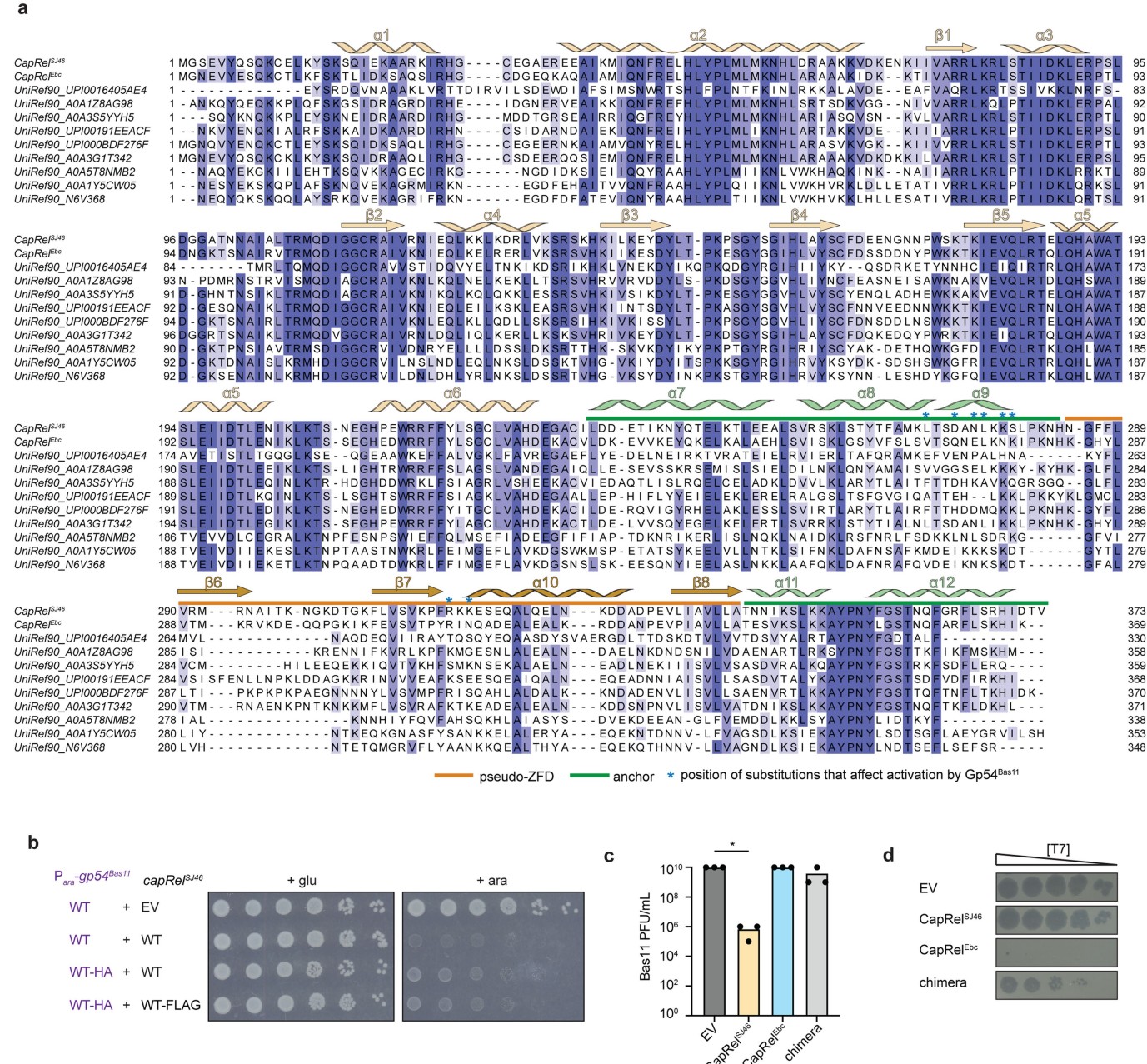

**Extended Data Fig. 2 | Gp54^Bas11 binds directly to CapRel^SJ46. (a)** Multiple sequence alignment of CapRel^SJ46 and diverse CapRel homologs, with labels indicating the secondary structural elements of CapRel^SJ46 and the antitoxin domain (pseudo-ZFD and anchors). The residues whose substitutions affect the activation of CapRel^SJ46 by Gp54^Bas11 are highlighted with stars. Alignment was adapted from previous work[11]. **(b)** Serial dilutions of cells producing CapRel^SJ46 or a FLAG-tagged version from its native promoter and Gp54^Bas11 or a HA-tagged version from an arabinose-inducible promoter on media containing glucose or arabinose. **(c)** Summary of 3 independent replicates of phage spotting assay in Fig. 2d. Asterisk indicates $p = 10^{-18}$ (unpaired two-tailed t-test). **(d)** Serial dilutions of phage T7 spotted on lawns of cells harboring an empty vector (EV) or a plasmid producing CapRel^SJ46, CapRel^Ebc or the chimera. Data also shown in previous work[11].

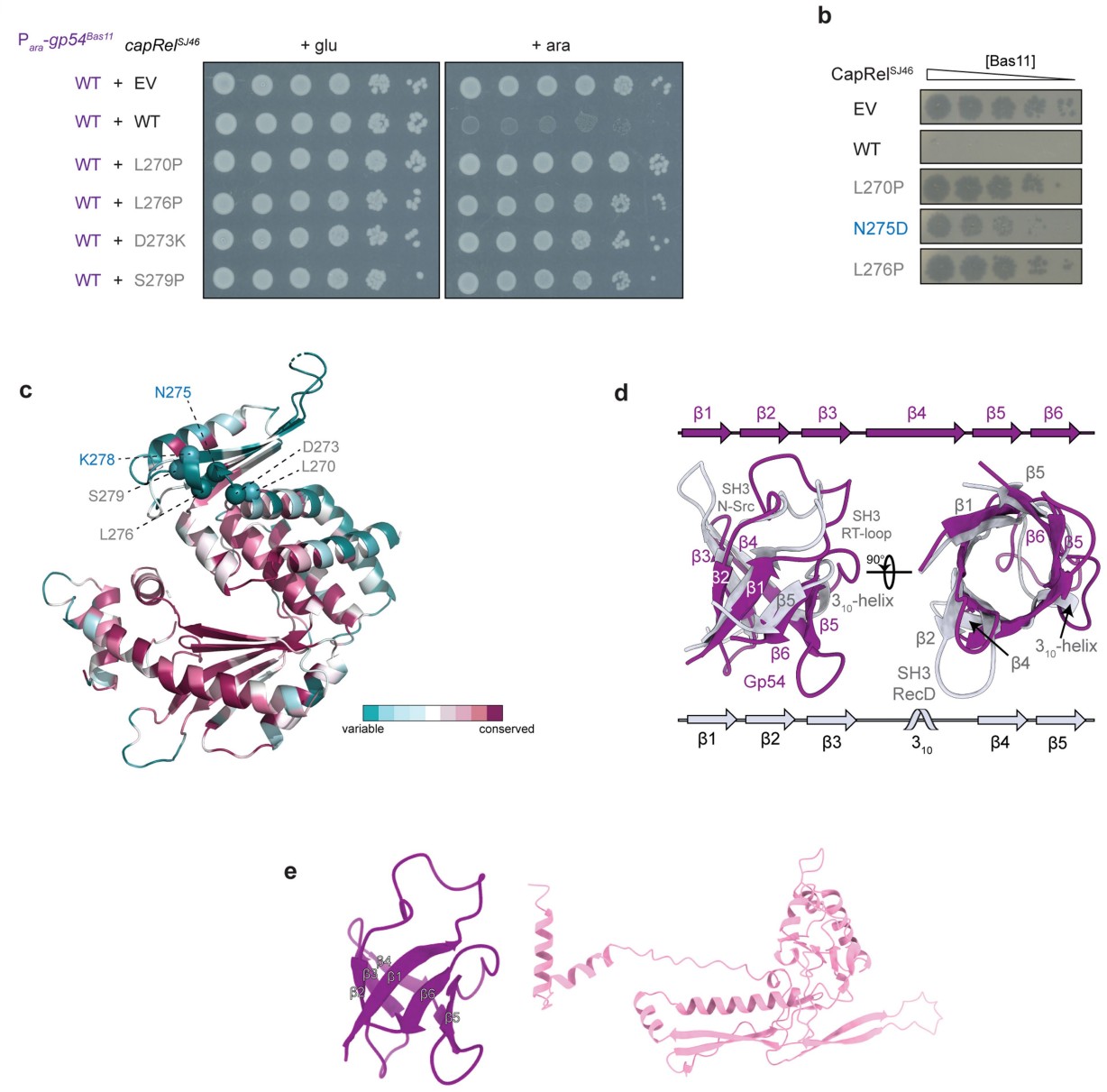

**Extended Data Fig. 3 | Structural analysis of Gp54[Bas11] and its interaction with CapRel[SJ46].** (**a**) Serial dilutions of cells producing the indicated variant of CapRel[SJ46] from its native promoter and the wild-type Gp54[Bas11] from an arabinose-inducible promoter on media containing glucose or arabinose. (**b**) Serial dilutions of phage Bas11 spotted on lawns of cells harboring an empty vector (EV) or a plasmid producing CapRel[SJ46] or the indicated variant. (**c**) Cartoon representation of the crystal structure of CapRel[SJ46] colored by the conservation score of each amino acid calculated by ConSurf[49].

Substitutions in the α-helix (α9) formed by residues 270-279 are labeled as spheres. (**d**) Comparison of Gp54[Bas11] with the 5 β-stranded β-barrel SH3 domain (residues 470 to 533) from the RecD subunit of the RecBCD repair complex (PDB 5LD2[50]). The two domains superimposed with a Z-score of 3.8. The secondary structural elements of both proteins are labeled. (**e**) Cartoon representation of the crystal structure of Gp54[Bas11] (*left*) and the AlphaFold-predicted structure of MCP[SECΦ27] (*right*).

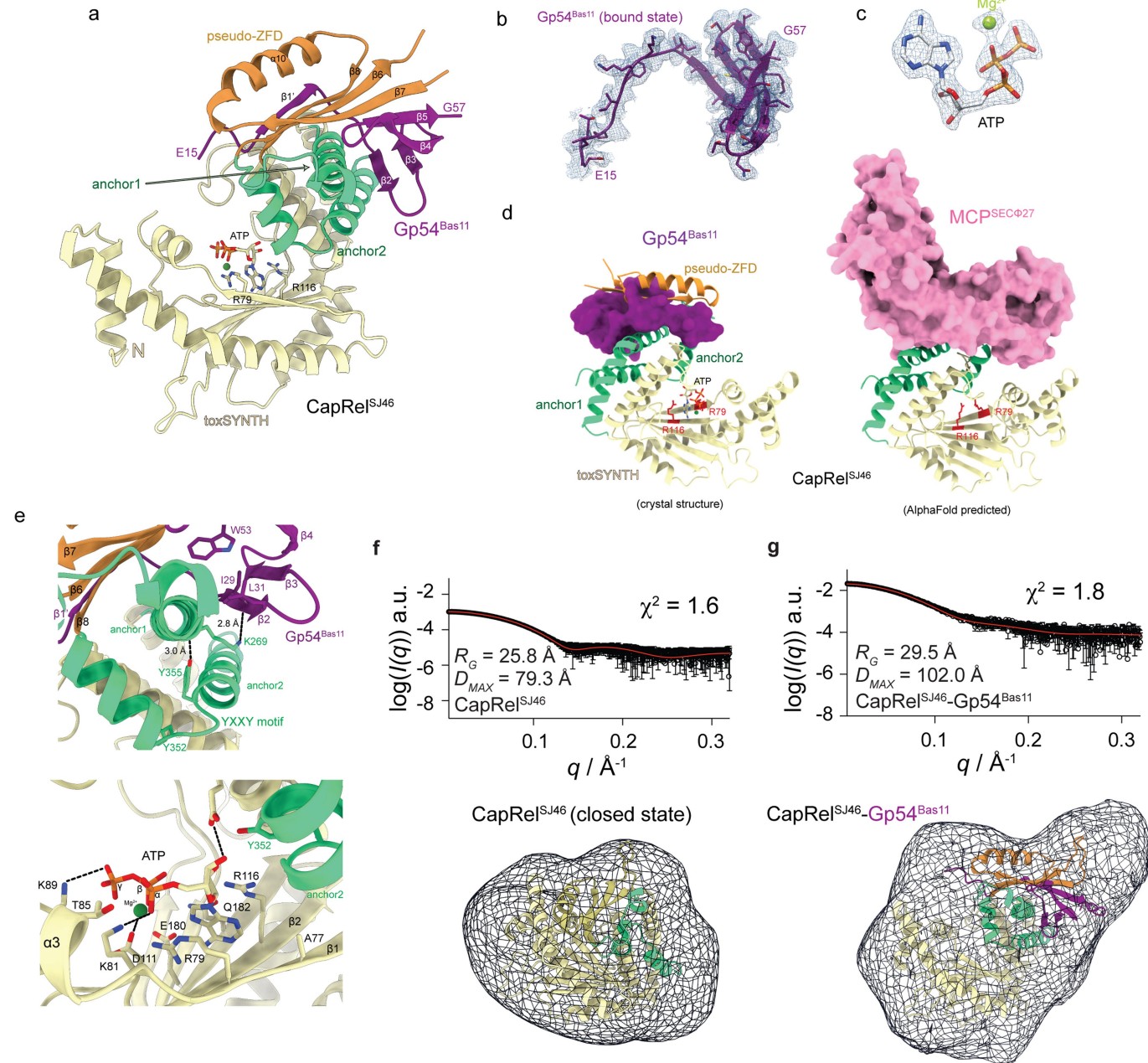

**Extended Data Fig. 4 | Structural analysis of the CapRel^SJ46-Gp54^Bas11 complex.**
(**a**) Cartoon representation of the crystal structure of CapRel^SJ46-Gp54^Bas11
complex bound to ATP. ATP coordination residues of CapRel^SJ46 toxin domain
(R79 and R116) are labeled. (**b**) Cartoon representation and the corresponding
m2Fo-DFc electron density map of Gp54^Bas11 in the CapRel^SJ46-bound state.
(**c**) The unbiased mFo-DFc electron density map of ATP and Mg^2+ observed in
the crystal structure of CapRel^SJ46-Gp54^Bas11 complex. (**d**) *Left*, crystal structure
of the complex of Gp54^Bas11 (purple) bound to CapRel^SJ46 (colored by domains).
*Right*, predicted structural model of the complex of CapRel^SJ46 and MCP^SECΦ27
(pink) by AlphaFold. A different view of Fig. 3c. (**e**) Detailed interface of
CapRel^SJ46-Gp54^Bas11 complex structure indicating interaction with Gp54^Bas11

via K269 (*top*) tethers Y355 of the YXXY neutralization motif away from the
toxin active site of CapRel^SJ46 (*bottom*). (**f**) *Top*, experimental SAXS analysis
of CapRel^SJ46 (open black circles). The theoretical scattering of CapRel^SJ46 in
the closed state predicted by AlphaFold is shown in red. *Bottom*, comparison
of the model of CapRel^SJ46 in the unbound closed state with an ab-initio
envelope calculated from the experimental SAXS data using DAMMIF[48].
(**g**) *Top*, experimental SAXS analysis of CapRel^SJ46-Gp54^Bas11 complex (open
black circles). The theoretical scattering of the CapRel^SJ46-Gp54^Bas11 crystal
structure is shown in red. *Bottom*, comparison of the crystal structure of
CapRel^SJ46-Gp54^Bas11 with an ab-initio envelope calculated from the
experimental SAXS data using DAMMIF[48].

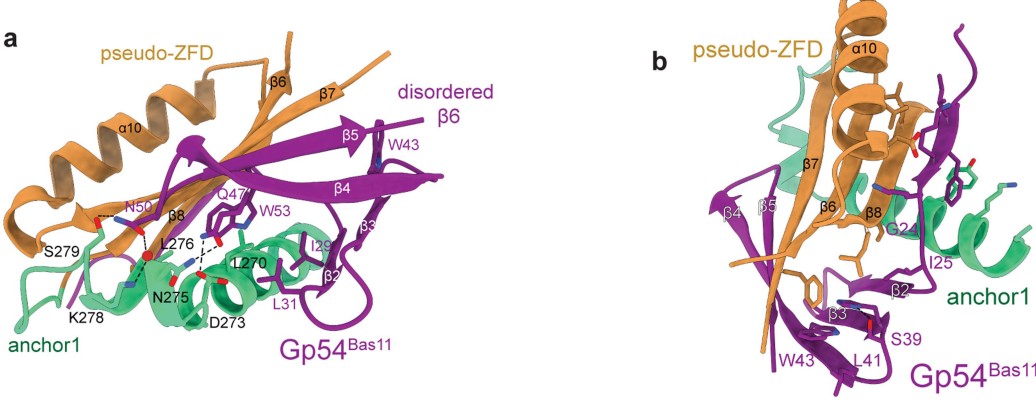

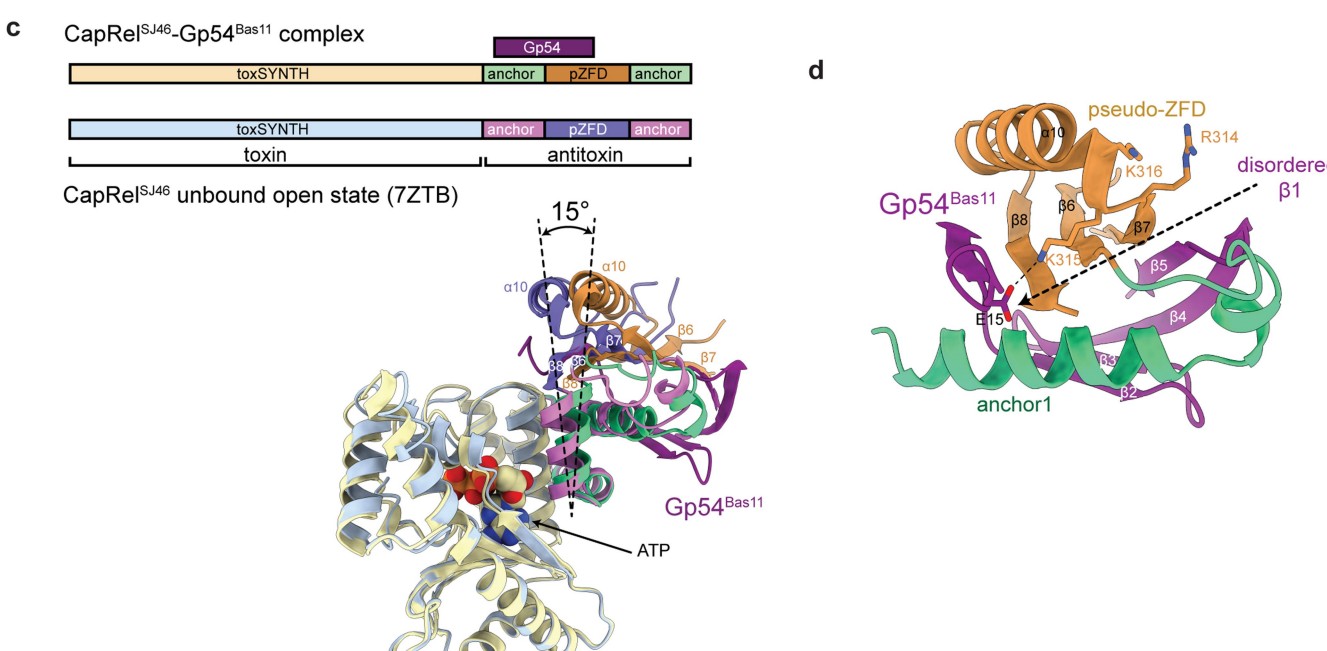

**Extended Data Fig. 5 | Gp54^Bas11 interacts with the antitoxin region of CapRel^SJ46.** (**a**) Details of the interface formed by the antitoxin domain of CapRel^SJ46 and Gp54^Bas11 observed in the crystal structure. Residues in anchor-1 of CapRel^SJ46 and residues in Gp54^Bas11 involved in the interface are labeled. (**b**) Detailed interface of Gp54^Bas11 bound to CapRel^SJ46. Residues in Gp54^Bas11 that are involved in hydrophobic interaction with CapRel^SJ46 are labeled.

(**c**) Superposition and comparison of CapRel^SJ46 in complex with Gp54^Bas11 and unbound open state (PDB 7ZTB). (**d**) Details of the interface of CapRel^SJ46 bound to Gp54^Bas11, with residues 314-316 of CapRel^SJ46 labeled. The most N-terminal residue observed in crystal structure (E15) of Gp54^Bas11 and the disordered β1 are indicated.

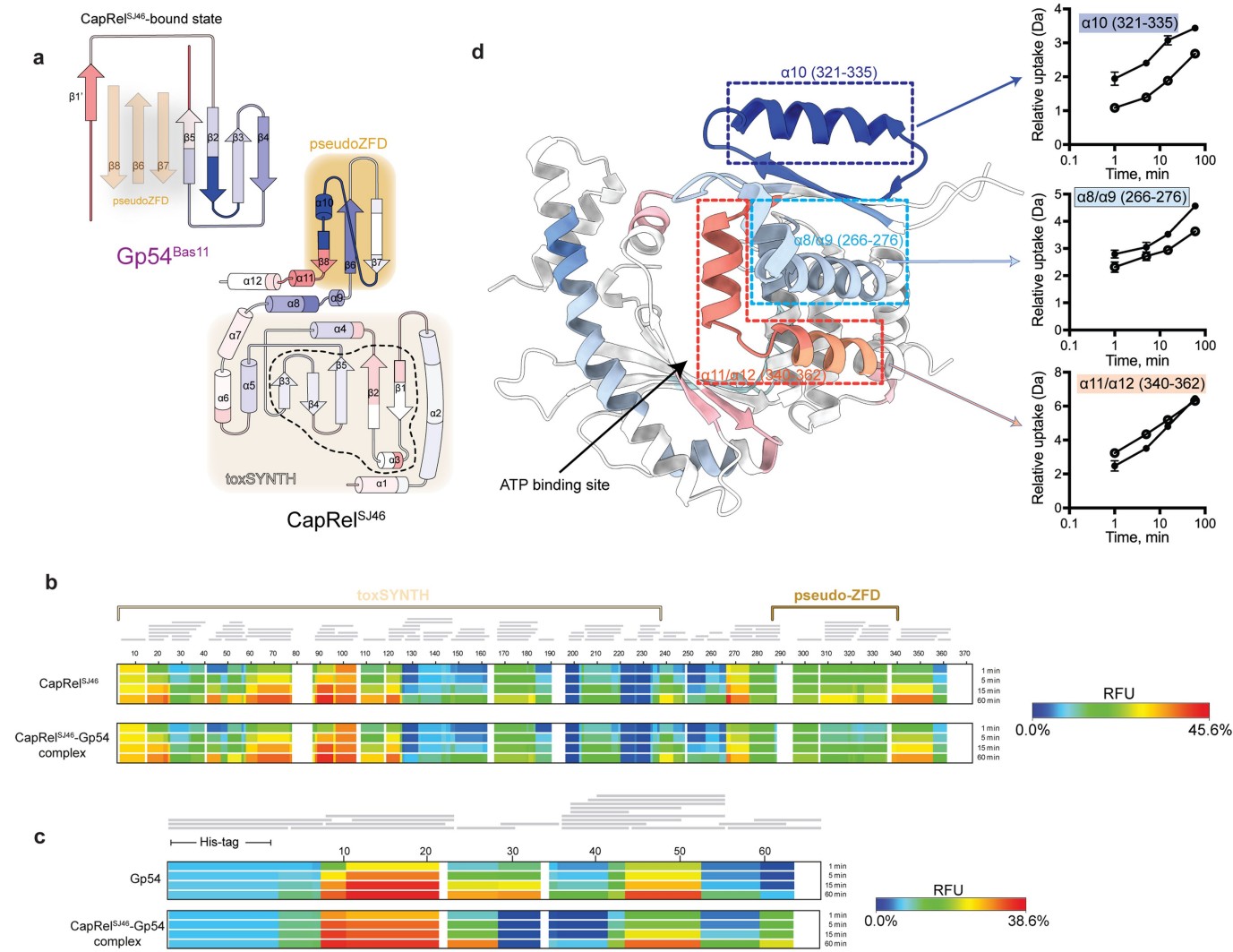

**Extended Data Fig. 6 | Characterization of the CapRel^SJ46-Gp54^Bas11 and CapRel^SJ46-MCP^SECΦ27 interactions.** (**a**) Topological representation of CapRel^SJ46 and Gp54^Bas11 colored according to the ΔHDX as in Fig. 3f,g. (**b**) Heat maps representing the HDX of CapRel^SJ46 (*top*) and CapRel^SJ46-Gp54^Bas11 complex (*bottom*). The relative fractional uptake (RFU) is indicated based on the color scale shown. (**c**) Heat maps representing the HDX of Gp54^Bas11 (*top*) and CapRel^SJ46-Gp54^Bas11 complex (*bottom*). The relative fractional uptake (RFU) is indicated based on the color scale shown. (**d**) *Left*, ΔHDX after 5 min, between CapRel^SJ46 and the CapRel^SJ46-Gp54^Bas11 complex plotted as a heat map on the structure of CapRel^SJ46 in the open, active state. *Right*, evolution of the deuterium uptake kinetics of peptides from α8/α9 (266-276), α10 (321-335), and α11/α12 (340-362) of CapRel^SJ46 in the unbound (solid black circles) and Gp54^Bas11-bound (open black circles) states. Measurements were performed in triplicate for each time point in each condition. Error bars indicate standard deviation (n = 3).

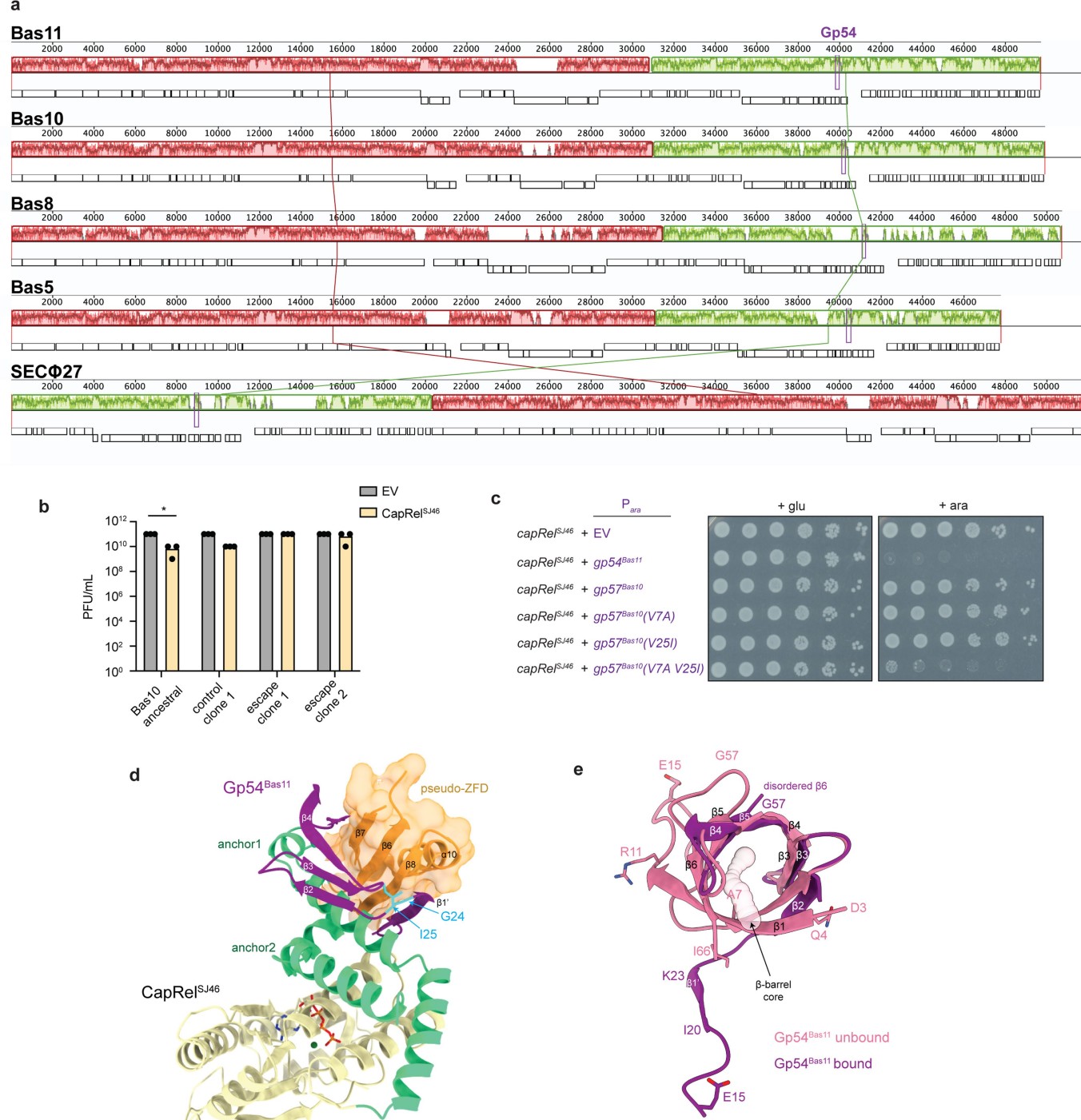

**Extended Data Fig. 7 | Analysis of Gp54 homologs in related phages.**
(**a**) Whole genome alignment of phages Bas11, Bas10, Bas8, Bas5 and SECΦ27. Gp54 and its homologs are labeled with purple boxes. Green and red colored blocks represent corresponding homologous regions of phages that had genome rearrangement, with the height of these blocks indicating similarities. (**b**) Summary of 3 independent replicates of phage spotting assay in Fig. 4b. Asterisk indicates p = 10⁻⁶ (unpaired two-tailed t-test). (**c**) Serial dilutions of cells producing CapRel$^{SJ46}$ from its native promoter and Gp54$^{Bas11}$ or Gp57$^{Bas10}$

(the Gp54 homolog in Bas10) or the indicated variant from an arabinose-inducible promoter on media containing glucose or arabinose. (**d**) The crystal structure of Gp54$^{Bas11}$ (purple) bound to CapRel$^{SJ46}$ (colored by domains). Substitutions in Gp54$^{Bas11}$ observed in Bas11 escape mutant (G24D) and residue I25 that is different between Gp54$^{Bas11}$ and Gp57$^{Bas10}$ are colored in cyan. (**e**) Comparison of the unbound (pink) and the CapRel$^{SJ46}$-bound state (purple) of Gp54$^{Bas11}$, indicating the structural changes. Residue A7V might stabilize the β-barrel core in the unbound state.

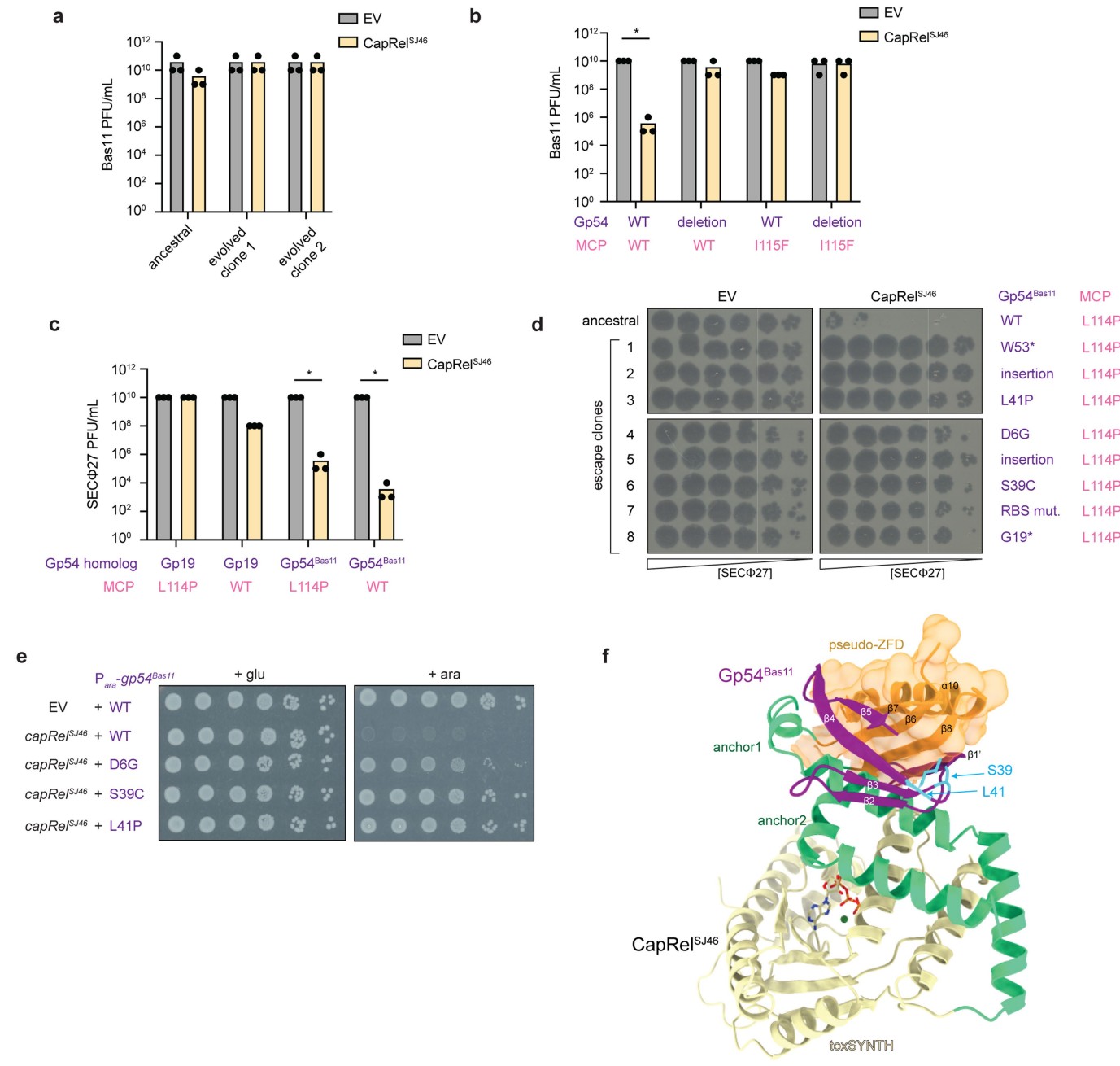

**Extended Data Fig. 8 | Both Gp54^Bas11 and MCP activate CapRel^SJ46 in Bas11 phage.** (**a**) Summary of 3 independent replicates of phage spotting assay in Fig. 5a. (**b**) Summary of 3 independent replicates of phage spotting assay in Fig. 5b. Asterisk indicates p = 10^−18 (unpaired two-tailed t-test). (**c**) Summary of 3 independent replicates of phage spotting assay in Fig. 5c. Asterisks indicate p = 10^−18 (Gp54^Bas11 with MCP(L114P)) or 10^−26 (Gp54^Bas11 with MCP) (unpaired two-tailed t-test). (**d**) Serial dilutions of the ancestral SECΦ27 MCP(L114P) phage harboring Gp54^Bas11 and eight escape clones spotted on lawns of cells

harboring an empty vector (EV) or a plasmid producing CapRel^SJ46. The corresponding genotypes of its MCP and Gp54^Bas11 are indicated on the right. Three independent replicates are shown in Fig. 5d. (**e**) Serial dilutions of cells producing CapRel^SJ46 from its native promoter and the indicated variant of Gp54^Bas11 from an arabinose-inducible promoter on media containing glucose or arabinose. (**f**) The crystal structure of Gp54^Bas11 (purple) bound to CapRel^SJ46 (colored by domains). Residues S39 and L41 in Gp54^Bas11 substituted in the SECΦ27 MCP(L114P) escape mutants are colored in cyan.

**Extended Data Table 1 | X-ray data collection and refinement statistics**

| | Gp54<sup>Bas11</sup> | CapRel<sup>SJ46</sup>-Gp54<sup>Bas11</sup> complex |
|---|---|---|
| Wavelength (Å) | 1.5406 | 0.9999 |
| Resolution range (Å) | 24.29 - 2.30 (2.38 - 2.30) | 50.128 - 2.25 (2.36 - 2.25) |
| Space group | P1 | C222$_1$ |
| Unit cell | | |
| *a, b, c* | 50.0, 54.6, 54.6 | 49.7, 103.3, 209.2 |
| α, β, γ | 78.5, 78.8, 69.9 | 90.0, 90.0, 90.0 |
| Total reflections | 42837 (4144) | 47938 (4615) |
| Unique reflections | 21642 (2168) | 22581 (1611) |
| Multiplicity | 2.0 (1.9) | 9.4 (5.6) |
| Completeness (%) | 92.4 (91.1) | 90.5 (72.2) |
| Mean I/sigma(I) | 8.4 (1.8) | 9.1 (1.3) |
| Wilson B-factor | 34.6 | 34.7 |
| R-merge | 0.06 (0.40) | 0.27 (1.39) |
| R-pim | 0.06 (0.40) | 0.09 (0.64) |
| CC1/2 | 0.995 (0.811) | 0.987 (0.29) |
| R-work (%) | 21.1 (28.1) | 19.3 (27.9) |
| R-free (%) | 25.2 (35.3) | 23.6 (31.7) |
| CC(work) | 0.944 (0.856) | 0.932 (0.600) |
| CC(free) | 0.919 (0.788) | 0.911 (0.528) |
| Number of non-hydrogen atoms | 3245 | 3487 |
| macromolecules | 3110 | 3094 |
| ligands | 0 | 39 |
| solvent | 135 | 354 |
| RMS deviations | | |
| Bond lengths (Å) | 0.002 | 0.014 |
| Bond angles (°) | 0.48 | 1.49 |
| Ramachandran favored (%) | 100.0 | 98.4 |
| Ramachandran allowed (%) | 0.0 | 1.6 |
| Ramachandran outliers (%) | 0.0 | 0.0 |
| Rotamer outliers (%) | 2.0 | 5.1 |
| Clashscore | 4.0 | 6.0 |
| Average B-factor | 45.7 | 32.0 |
| macromolecules | 46.0 | 31.0 |
| ligands | - | 28.7 |
| solvent | 39.1 | 41.3 |
| PDB ID | 9AXB | 9ERV |

Statistics for the highest-resolution shell are shown in parentheses.

**Extended Data Table 2 | Comparison between the SAXS experimental scattering curve and the theoretical scattering derived from structural models of each species**

| Titrations | $R_g$ (Å) | $D_{max}$ (Å) | fitting $\chi^2$ to experimental data |
|---|---|---|---|
| CapRel[SJ46] (closed state) | 25.8 | 79.3 | 1.6 |
| CapRel[SJ46] (open state) | - | - | 5.7 |
| CapRel[SJ46] - Gp54[Bas11] complex | 29.9 | 102.0 | 1.8 |

# Reporting Summary

## Statistics

For all statistical analyses, confirm that the following items are present in the figure legend, table legend, main text, or Methods section.

| n/a | Confirmed | |
|---|---|---|
| ☐ | ☒ | The exact sample size (*n*) for each experimental group/condition, given as a discrete number and unit of measurement |
| ☐ | ☒ | A statement on whether measurements were taken from distinct samples or whether the same sample was measured repeatedly |
| ☐ | ☒ | The statistical test(s) used AND whether they are one- or two-sided *Only common tests should be described solely by name; describe more complex techniques in the Methods section.* |
| ☒ | ☐ | A description of all covariates tested |
| ☒ | ☐ | A description of any assumptions or corrections, such as tests of normality and adjustment for multiple comparisons |
| ☐ | ☒ | A full description of the statistical parameters including central tendency (e.g. means) or other basic estimates (e.g. regression coefficient) AND variation (e.g. standard deviation) or associated estimates of uncertainty (e.g. confidence intervals) |
| ☐ | ☒ | For null hypothesis testing, the test statistic (e.g. *F*, *t*, *r*) with confidence intervals, effect sizes, degrees of freedom and *P* value noted *Give P values as exact values whenever suitable.* |
| ☒ | ☐ | For Bayesian analysis, information on the choice of priors and Markov chain Monte Carlo settings |
| ☒ | ☐ | For hierarchical and complex designs, identification of the appropriate level for tests and full reporting of outcomes |
| ☒ | ☐ | Estimates of effect sizes (e.g. Cohen's *d*, Pearson's *r*), indicating how they were calculated |

*Our web collection on statistics for biologists contains articles on many of the points above.*

## Software and code

Policy information about availability of computer code

| Data collection | NCBI BLAST for homology searches. Consurf 2016 web server for identifying additional homologs. XDS suite: Jun 30, 2023 (BUILT 20230630) |
|---|---|
| Data analysis | Coot: 0.9.8.93 for crystallography. Phenix: 1.20.1-4487 for crystallography. Buster/TNT: 2.10.4 for crystallography. UCSF ChimeraX version 1.7rc202311230013 for structural visualization. Geneious 2022.0.2 for genome and sequence alignment. |

For manuscripts utilizing custom algorithms or software that are central to the research but not yet described in published literature, software must be made available to editors and reviewers. We strongly encourage code deposition in a community repository (e.g. GitHub). See the Nature Portfolio guidelines for submitting code & software for further information.

nature portfolio | reporting summary

## Data

Policy information about availability of data

All manuscripts must include a data availability statement. This statement should provide the following information, where applicable:

- Accession codes, unique identifiers, or web links for publicly available datasets
- A description of any restrictions on data availability
- For clinical datasets or third party data, please ensure that the statement adheres to our policy

Structural data for Gp54 and the CapRelSJ46-Gp54 complex are available in PDB (9AXB and 9ERV, respectively). Sequencing data are available in the Sequence Read Archive (SRA) under BioProject PRJNA1084025. HDX-MS data can be accessed via ProteomeXchange with identifier PXD050526. AlphaFold predicted structural models are deposited in the ModelArchive Database (https://www.modelarchive.org) with the accession codes ma-zblch (DOI:10.5452/ma-zblch) and ma-9z23e (DOI:10.5452/ma-9z23e). All other data are available in the manuscript or the supplementary materials. Other previously published structures are available in PDB (7ZTB, 5LD2). UniRef90 database is publicly available. Reference phage genomes are publicly available: SECΦ27 (NC_047938.1), Bas05 (MZ501101.1), Bas08 (MZ501059.1), Bas10 (MZ501077.1), Bas11 (MZ501085.1). Materials including strains and plasmids are available upon reasonable request.

## Research involving human participants, their data, or biological material

Policy information about studies with human participants or human data. See also policy information about sex, gender (identity/presentation), and sexual orientation and race, ethnicity and racism.

| | |
|---|---|
| Reporting on sex and gender | N/A |
| Reporting on race, ethnicity, or other socially relevant groupings | N/A |
| Population characteristics | N/A |
| Recruitment | N/A |
| Ethics oversight | N/A |

Note that full information on the approval of the study protocol must also be provided in the manuscript.

# Field-specific reporting

Please select the one below that is the best fit for your research. If you are not sure, read the appropriate sections before making your selection.

☒ Life sciences ☐ Behavioural & social sciences ☐ Ecological, evolutionary & environmental sciences

For a reference copy of the document with all sections, see nature.com/documents/nr-reporting-summary-flat.pdf

# Life sciences study design

All studies must disclose on these points even when the disclosure is negative.

| | |
|---|---|
| Sample size | No sample size calculation was performed. Sample sizes were chosen based on the number needed to reliably determine differences between groups. Given large effect sizes, we chose to replicate experiments 2-3 times as is routine to simply indicate reproducibility. |
| Data exclusions | No data were excluded from analysis. |
| Replication | All experimental findings were repeated at least twice. All reported results were successfully reproduced. |
| Randomization | No experimental groups or control groups were subjectively chosen and there are no covariates to control for as experiments were done in isogenic strains. No randomization is required. |
| Blinding | Blinding was not required because all data were obtained objectively and had strong effect sizes over multiple independent replicates and raw data are reported in the manuscript. |

# Reporting for specific materials, systems and methods

We require information from authors about some types of materials, experimental systems and methods used in many studies. Here, indicate whether each material, system or method listed is relevant to your study. If you are not sure if a list item applies to your research, read the appropriate section before selecting a response.

## Materials & experimental systems

| n/a | Involved in the study |
|-----|----------------------|
| ☐ | ☒ Antibodies |
| ☒ | ☐ Eukaryotic cell lines |
| ☒ | ☐ Palaeontology and archaeology |
| ☒ | ☐ Animals and other organisms |
| ☒ | ☐ Clinical data |
| ☒ | ☐ Dual use research of concern |
| ☒ | ☐ Plants |

## Methods

| n/a | Involved in the study |
|-----|----------------------|
| ☒ | ☐ ChIP-seq |
| ☒ | ☐ Flow cytometry |
| ☒ | ☐ MRI-based neuroimaging |

## Antibodies

| | |
|---|---|
| Antibodies used | HA-tag (C29F4) rabbit mAb (Cell Signaling Technology #3724)<br>(FLAG) DYKDDDDK Tag (D6W5B) Rabbit mAb (Cell Signaling Technology, #14793)<br>Anti-RpoA antibody (Biolegend Cat#: 663104). |
| Validation | Antibodies have been validated based on manufacturer's website:<br>anti-HA: validated by "western blot analysis of extracts from HeLa cells, untransfected or transfected with either HA-FoxO4 or HA-Akt3".<br>anti-FLAG: validated by "western blot analysis of extracts from 293T cells, mock transfected (-) or transfected with DYKDDDDK-GFP".<br>In addition, all western blotting experiments include controls with untagged proteins as internal validation for antibodies.<br>anti-RpoA: validated by western blots using "total lysates from HeLa and E.coli BL21 cells" by the manufacturer, and widely used in bacterial studies.<br>All antibodies were used according to the manufacturer's guidelines. |

## Plants

| | |
|---|---|
| Seed stocks | N/A |
| Novel plant genotypes | N/A |
| Authentication | N/A |

