## [Peer Review File · Nature]

Manuscript Title: A bacterial immunity protein directly senses two disparate phage proteins

Reviewer Comments & Author Rebuttals

Reviewer Reports on the Initial Version:

Referees' comments:

Referee #1 (Remarks to the Author):

In this study Zhang and colleagues discovered a new trigger for the anti-phage defense protein CapRelSJ46, a toxin-antitoxin system previously described to bind major capsid proteins (MCP). This newly discovered trigger Gp54 of Bas11 is unrelated to MCPs. The authors solved the structure of Gp54Bas11 and Gp54Bas11 in complex with CapRelSJ46, showing great conformational changes in Gp54 upon binding to CapRelSJ46. This protein-protein interface overlaps but is different from the predicted interface of the CapRelSJ46-MCPSECphi27 complex. Phage Bas11 that harbours both triggers must mutate both to evade CapRelSJ46 defense. This significantly complicates the phage's ability to escape detection. Overall, the authors propose a summary where CapRelSJ46 can sense two completely unrelated phage proteins using the same sensory domain but with distinct interfaces. This dual recognition challenges the previously assumed one-to-one relationship between bacterial immunity proteins and their triggers, changing our understanding of the evolutionary arms race between bacteria and their viruses.

I really enjoyed reading this paper, which was beautifully written, with a clear message, supported by good data. The methodology is sound, appropriate and well presented. I do think this study will have a conceptual impact on how we think about what triggers immune systems in bacteria, and I enjoyed how the authors contextualized this more generally with immune systems in general. I believe it will be of interest for a general readership.

Major comments

Here I suggest some ideas that could add to the study. I don't feel the authors need to address them experimentally for the study to be valuable but this could be used to expand the significance of the findings.

- is it two triggers or more? Could the authors somehow perform a screen, or use in silico predictions to estimate if others and ideally how many proteins from e. coli phages (and even better from coli MGE / pan genomes) are potentially expected to bind to this protein / trigger it. This would deepen our understanding of the plasticity of this interface and have potentially important evolutionary consequences (cf below) if this number is large.

- An important point regarding the evolutionary dynamics at play that is currently not raised or addressed even in the discussion is the disadvantages of responding to two triggers. For me, the main cost would come from autoimmunity. With less selective proteins-protein recognition, the probability of autoimmunity would become much higher. What are the estimated costs of this? Could proteins from the host or present on for example incoming MGE trigger the system? Is there a

way to estimate when this would happen? To estimate cost, authors could run competition experiments even though it would only partially cover the source of potential costs, which from me could quickly arise upon HGT of novel genes.

- Another big question is how general this phenomenon is. This is difficult to address, but there might be immune systems where this is already suspected which could then be verified (for example PARIS).

- I also wondered, would these types of evolutionary pressures lead to specific structural properties for the sensor proteins? What could these be? Would be interesting to read the authors speculate on this as I think it would be quite difficult to think and test for this.

- What is the role of Gp54? I understand this falls probably out of the scope of the study but would be interesting to learn more

Minor comments:

- Figures should be better labelled, for example, a table next to the spot assays will make it easier for the reader to follow which proteins are expressed and which version of the protein is used. This would help the reader in figures 1d, 2b,2e, 3a, 3g, 4b-e, 5a-b. In figures 1a, 2c, 2d, S5a,b,e the labels of the domains are barely visible, they could be put as the labels of Figure S4a.
- In Fig2b and Fig3h the bands for anti-HA of Gp54 are barely visible, especially in the lysate, authors might consider trying to further optimize the gel and blotting conditions. Besides, loading control missing for Fig2b and Fig3h, authors could add it as source data or supplementary.
- showing a comparison of the genome of the different phages used in this study would be a nice add.
- the concept of Red Queen dynamic could be introduced earlier in the text
- Give maps/models to the reviewers.
- Deposit AF model in a database (like modelarchive) and add how the models were obtained in the methods section.
- The authors could follow Gp54 mRNA levels by RT-qPCR in the clones with mutations upstream of the gene.
- The authors should state in the text when the structure is predicted and not solved, for example, in line 218: add “predicted” before “interface”, line 502: add “predicted” before “structural model”.
- Circular dichroism not included in the methods section.
- Line 366: “completely” written twice in the same sentence.
- Reorganize the panels of Fig 3.
- In Fig3d show as atoms R314 and K316 if they are highlighted in the left side of the panel.
- In the legend of Fig5f, a longer description of the proposed mechanism would help the readers.
- The topology could be added to the alignment in FigS2a.
- In FigS7c and S8f the highlight in red of the ATP site could be removed to stress out the mutations pointed.

Referee #2 (Remarks to the Author):

A. Summary of the key results

The authors identified new (second) activator of CapRel SJ46 bacterial immunity, which binds the same antitoxin domain as the previously discovered activator (and they identified specific residues in CapRel that were involved). They then present multiple crystal structures of the Gp54 ligand alone and in complex with CapRel SJ46, showing a very unusual 6-stranded beta barrel for the Gp54 protein, which dramatically refolds upon CapRel SJ46 binding. They confirmed these structures with specific substitution mutants and HDX mass spectrometry. These structures allowed the authors to create CapRel SJ46 mutants that disrupt Gp54 binding, but not the previously discovered activator MCP.

In Fig 4 the authors return to the Basel collection and testing whether homologs of the two activators, MCP and Gp54, also activated CapRel. One of the more notable findings from this section was the differences between the Gp54 phenotypes from Bas10 and Bas11, which differed only in three amino acids, two of which abolished CapRel SJ46 activation. Because Bas10 and Bas11 had identical MCP but each strain selected for different kinds of escape mutants, the authors then returned to their Bas11 escapers, finding that they only partly escaped immune targeting. Full escape required both MCP and Gp54 mutations, as both were activators of CapRel SJ46. Further, when Gp54 Bas11 was engineered into the other, previously studied phage SECphi27, defense was stronger and no spontaneous mutants were observed.

B. Originality and significance

The authors devote much of their abstract, introduction, and discussion to the fact that CapRel SJ46 can be activated by two distinct phage triggers, and the subsequent implications. While I find this discovery to be interesting and elegant, for me it was not the most surprising part of their paper. I thought that the entropically driven binding involving such extensive protein unfolding and re-folding seemed wild and cool. Admittedly, this is more outside my area of expertise, but if the authors agree with me, they may want to consider refocusing their framing.

So, how unexpected is multifactorial sensing by bacterial defense systems? As the authors state on lines 53-54, for many bacterial immunity proteins, no triggers have been discovered. It is therefore not surprising that no one had previously discovered multiple triggers for a given protein. It is still interesting and cool that the authors have discovered multifactorial sensing here, and very nice that it prevents spontaneous escape mutants from arising, as you would expect. But I would classify that more as a 'gap in knowledge that is now filled' rather than as a large paradigm shift.

In addition, parallel phenomena have been previously described in animal antiviral immunity. I recommend that in lines 41-48, the authors also discuss and cite eukaryotic immune proteins that can bind different, nonhomologous viral proteins using similar but non-identical molecular interfaces, such as the case of MxA (e.g. doi: 10.1074/jbc.M113.543892, 10.1051/vcm/2022002).

C. Data & methodology: validity of approach, quality of data, quality of presentation

The approaches and experiments were the right ones to do and the data was of high quality. Most of the figures were well-presented, although I found Fig 3 to be somewhat difficult to decode, which de-emphasized the substantial and cool refolding of Gp54 in the complex. I suggest that moving the AlphaFold predictions of the MCP complex to the supplement might simplify this figure and make its impact easier to see.

In Fig 5f&g, while I appreciated that the authors were attempting to provide figures that were not photos of spot dilutions, I did not find these figures to be useful. It took a while to decode what the authors were trying to say in these cartoons, and once I figured it out I found it was a message I had already gotten (more clearly) from other parts of the manuscript. Recommend redesigning or dropping these panels.

D. Appropriate use of statistics and treatment of uncertainties

Many of the datapoints are provided as pictures of spot dilutions and phage plaques, instead of quantification. This is standard in the field and it is nice to see the plaque morphology directly, as this has more information than simple PFU numbers. I do think it would be nice to report how much the plaquing efficiencies varied across the three independent replicates of each experiment that the authors state they performed. Especially in cases where the phenotypes are subtle, I would encourage the authors to report these numbers, perhaps adjacent to the images of the plates or in the supplement if necessary.

E. Conclusions: robustness, validity, reliability

The evidence reported appears to be strong and is backed up by multiple, independent lines of inquiry. Although see D above.

F. Suggested improvements: experiments, data for possible revision

I do not suggest any additional experiments. My suggestions are limited to those noted elsewhere, specifically:

- 1) Consider a re-framing of the significance of the manuscript (or re-write these sections to convince readers that multifactorial sensing is as surprising as stated)
- 2) Discuss parallels and/or differences to MxA, and potentially other animal immune genes
- 3) Revise the figure layout and presentation in Fig 3 & 5
- 4) Provide some quantification of the various spot dilution pictures, so that readers can appreciate the variation between replicates.

G. References: appropriate credit to previous work?

The authors are well-versed in the bacterial immunity literature and seem to cite previous work appropriately. I believe they can improve the contextualization of their work with respect to the animal immunity literature.

H. Clarity and context: lucidity of abstract/summary, appropriateness of abstract, introduction and conclusions

-The manuscript is very clear, logical, and well written. The reader is taken along through each step of the data, making it easy to see how and why the conclusions were drawn. I particularly thank the authors for spelling out the conundrum they faced in lines 275-277, which I found very clarifying for

the experiments in Fig 5.

-If the authors found co-expression of WT Gp54 and CapRel was toxic in Fig 1, how were they able to co-express them for IPs in Fig 2? I must have missed something here. Would appreciate clarification.

Referee #3 (Remarks to the Author):

Zhang et al. present interesting and timely results that reveal a single prokaryotic PRR can effectively sense two completely different PAMPs. The work is of high significance as it changes the 1:1 (PRR:PAMP) paradigm currently held in the field of prokaryote immunity. The combination of *in vivo*, structural, and biophysical techniques to thoroughly probe the hypothesis is well executed and strongly supports the conclusions made. The writing and figures are of excellent quality. To do my job as a reviewer well, I scoured the manuscript, supplemental files, and figures to find anything that could improve the manuscript. Although I believe the manuscript is acceptable for publication in its current state, I found two areas that I believe could improve the manuscript – but they are minor.

1. It is clear from the figure legend of figure 3 that the the model of CapRelSJ46 and MCPSECØ27 was produced by AlphaFold2. However, the text does not indicate this. On lines 170 – 171, and 217-218, the AlphaFold2 model is mentioned in a way that could be interpreted as if it were determined by direct methods. It is recommended that the authors indicate that the model is from an AlphaFold2 prediction in the text in these areas to avoid confusion.
2. On line 365- 366 there is a statement that reads “Two, dual sensing makes it more difficult for the phage to completely evade defense completely unless both triggers are mutated.” I recommend removing the second “completely” to clarify the meaning of the sentence.

Author Rebuttals to Initial Comments:

We thank the reviewers for their enthusiasm and positive comments on our manuscript. Below we respond to each of the questions raised, indicating how the text and/or figures were revised accordingly.

Referee #1 (Remarks to the Author):

In this study Zhang and colleagues discovered a new trigger for the anti-phage defense protein CapRelSJ46, a toxin-antitoxin system previously described to bind major capsid proteins (MCP). This newly discovered trigger Gp54 of Bas11 is unrelated to MCPs. The authors solved the structure of Gp54Bas11 and Gp54Bas11 in complex with CapRelSJ46, showing great conformational changes in Gp54 upon binding to CapRelSJ46. This protein-protein interface overlaps but is different from the predicted interface of the CapRelSJ46-MCPSECphi27 complex. Phage Bas11 that harbours both triggers must mutate both to evade CapRelSJ46 defense. This significantly complicates the phage's ability to escape detection. Overall, the authors propose a summary where CapRelSJ46 can sense two completely unrelated phage proteins using the same sensory domain but with distinct interfaces. This dual recognition challenges the previously assumed one-to-one relationship between bacterial immunity proteins and their triggers, changing our understanding of the evolutionary arms race between bacteria and their viruses.

I really enjoyed reading this paper, which was beautifully written, with a clear message, supported by good data. The methodology is sound, appropriate and well presented. I do think this study will have a conceptual impact on how we think about what triggers immune systems in bacteria, and I enjoyed how the authors contextualized this more generally with immune systems in general. I believe it will be of interest for a general readership.

We thank the reviewer for their enthusiastic and supportive comments on our manuscript, and especially for noting the conceptual impact it will have on the field.

Major comments

Here I suggest some ideas that could add to the study. I don't feel the authors need to address them experimentally for the study to be valuable but this could be used to expand the significance of the findings.

- is it two triggers or more? Could the authors somehow perform a screen, or use in silico predictions to estimate if others and ideally how many proteins from e. coli phages (and even better from coli MGE / pan genomes) are potentially expected to bind to this protein / trigger it. This would deepen our understanding of the plasticity of this interface and have potentially important evolutionary consequences (cf below) if this number is large.

At this point we only have definitive data, particularly biochemical and structural data, to support the conclusion that CapRel^{SJ46} has two triggers and both function as triggers in the context of phage infection. However, we previously (Zhang et al 2022) reported that CapRel^{SJ46} robustly protects *E. coli* against infection by T-even phages that (i) don't encode a Gp54

homolog and (ii) have very different capsid proteins compared to SEC ϕ 27 and Bas11. So we think there may well be additional triggers, further underscoring the remarkable versatility of this immunity protein, but we feel it's overly speculative to include much on this in the manuscript other than a short discussion point (lines 367-369). We have considered using in silico (e.g. AlphaFold-based) predictions to identify this T-even-based trigger and potentially other triggers, but there are two important considerations. First, any such predictions are just that, predictions, and would need to be carefully studied in future work to determine their legitimacy as triggers, particularly in the context of phage infection. Second, although AlphaFold is immensely powerful, it isn't perfect either. Notably, given that Gp54 changes conformation dramatically upon binding to CapRel^{SJ46}, the AlphaFold-based prediction of Gp54 binding to CapRel^{SJ46} did not recapitulate the unusual, entropically-driven binding seen in our co-crystal structure.

- An important point regarding the evolutionary dynamics at play that is currently not raised or addressed even in the discussion is the disadvantages of responding to two triggers. For me, the main cost would come from autoimmunity. With less selective proteins-protein recognition, the probability of autoimmunity would become much higher. What are the estimated costs of this? Could proteins from the host or present on for example incoming MGE trigger the system? Is there a way to estimate when this would happen? To estimate cost, authors could run competition experiments even though it would only partially cover the source of potential costs, which from me could quickly arise upon HGT of novel genes.

This is indeed an important point. Autoimmunity could arise for any abortive infection system that is triggered by a phage protein, and potentially more so for systems like CapRel^{SJ46} that recognize multiple phage proteins. That said, it is notable that CapRel^{SJ46} is triggered by two proteins that do not have any remote homologs encoded by the host. In agreement with that, we have not seen any growth defects for cells carrying CapRel^{SJ46}. It could, however, require much longer timescales for a defect to emerge or, as the reviewer astutely notes, the HGT of novel genes. We now comment on this general notion of autoimmunity in the revised Discussion (lines 370-375).

- Another big question is how general this phenomenon is. This is difficult to address, but there might be immune systems where this is already suspected which could then be verified (for example PARIS).

This is a great question and while it is still early days for the field, we think there are a number of hints already that this phenomenon of multi-factorial sensing will turn out to be common. There are specific cases, such as the PARIS system noted by the reviewer, in which there are multiple proteins that are sufficient to activate and for which escape mutants have been found, although whether these proteins bind directly and whether they bind in different ways as with the MCP and Gp54 to CapRel remains to be shown. There is also the large-scale overexpression screen reported previously by the Typas lab (Bobonis et al 2022) which found that multiple genes from a single prophage were sometimes capable of triggering retron defense systems, though they have not yet determined whether they act directly and whether each is, in fact,

involved in triggering retrons during an infection. Finally, there is another large-scale screen from the Sorek lab (Stokar-Avihail et al 2023) that identified escape mutants in diverse phages that enable them to overcome a range of defense systems. Notably, many of the mutations isolated provide only partial escape. That paper speculated that such incomplete escape could be because individual mutations that provide full escape would inherently compromise the function that protein provides the phage. However, we think that in at least some cases it could instead reflect the presence of additional trigger proteins. We have now expanded our commentary on these other studies in our Discussion (lines 352-369) to provide readers with a stronger, more compelling sense of how common multi-factorial sensing is likely to prove.

- I also wondered, would these types of evolutionary pressures lead to specific structural properties for the sensor proteins? What could these be? Would be interesting to read the authors speculate on this as I think it would be quite difficult to think and test for this.

Indeed, at this point we can only speculate that anti-phage defense systems will rely on inherently plastic and versatile sensor domains like the pseudo-zinc finger domain (ZFD) in CapRel^{SJ46}. ZFDs are well known to bind a huge variety of other molecules, including both proteins and nucleic acids. However, as noted above, there could be auto-immunity issues that arise from using highly plastic sensor domains, so there may also be selective pressure to employ sensors that are unique or highly specific for individual phage triggers.

- What is the role of Gp54? I understand this falls probably out of the scope of the study but would be interesting to learn more.

We don't yet know but are actively exploring this question. As noted already in the manuscript (see lines 345-351), gene 54 is dispensable so we strongly suspect that it blocks a different defense system. Thus, Gp54 would potentially put the phage in a bind – it can either inhibit this other putative defense system but trigger CapRel^{SJ46}, or, if it mutates/loses gene 54 it would prevent CapRel^{SJ46} activation, but then be restricted by the other defense system. This would be analogous to what's been suggested for T7 Ocr protein, a small and non-essential protein like Gp54, that blocks RM systems while activating the PARIS system. We are currently trying to identify a defense system that Gp54 inhibits, but this is, as noted, well beyond the scope of this current study. We do, however, discuss the potential role of Gp54 in the Discussion - see lines 345-351.

Minor comments:

- Figures should be better labelled, for example, a table next to the spot assays will make it easier for the reader to follow which proteins are expressed and which version of the protein is used. This would help the reader in figures 1d, 2b,2e, 3a, 3g, 4b-e, 5a-b. In figures 1a, 2c, 2d, S5a,b,e the labels of the domains are barely visible, they could be put as the labels of Figure S4a.

We have tried to ensure that all spot assays have clear labels of what proteins are produced in each row and we have updated the labels of domains in the structural figures as suggested.

- In Fig2b and Fig3h the bands for anti-HA of Gp54 are barely visible, especially in the lysate, authors might consider trying to further optimize the gel and blotting conditions. Besides, loading control missing for Fig2b and Fig3h, authors could add it as source data or supplementary.

Although a bit dim due to the small size of Gp54, we do think these bands are still visible and the result of the co-IPs unambiguously clear and reproducible. We have repeated the anti-HA blots in lysates to improve visibility. We would also note that this is a co-IP not a straight immunoblot so there is no loading control for the IP samples. The blot for the bait protein indicates how much of it was pulled down and then the blot for the prey/co-IP'd protein shows how much of it was pulled down with the bait protein. Loading controls for the lysate samples (done by Coomassie stain of the membrane) have now been added to Supplementary Fig. 1 as suggested.

- showing a comparison of the genome of the different phages used in this study would be a nice add.

Thank you for this suggestion - we have now added such a comparison to Extended Data Fig. 7a.

- the concept of Red Queen dynamic could be introduced earlier in the text

We have now introduced this concept earlier, in the first paragraph of the main text (lines 48-53).

- Give maps/models to the reviewers.

Maps and models of crystal structures are deposited in PDB (9AXB, 9ERV) with the PDB validation reports provided as related manuscript files.

- Deposit AF model in a database (like modelarchive) and add how the models were obtained in the methods section.

AF models have been added to ModelArchive database as suggested. Accession numbers and how they were obtained are now included in the Methods section.

- The authors could follow Gp54 mRNA levels by RT-qPCR in the clones with mutations upstream of the gene.

Bas11 phage adsorbs very poorly to lab strains of *E. coli*, leading to asynchronous infection, which makes it difficult to track expression levels at a population level during infection. As an alternative, we cloned Gp54 harboring a FLAG epitope expressed from the wild-type promoter

or the mutated versions as observed in escape phage clones and used immunoblotting to assess levels. We observed clear differences that support our initial hypothesis. These data have now been added to Extended Data Fig. 1d and discussed on lines 103-104 of the revised text.

- The authors should state in the text when the structure is predicted and not solved, for example, in line 218: add “predicted” before “interface”, line 502: add “predicted” before “structural model”.

This has been updated as suggested (see lines 169, 178, 539).

- Circular dichroism not included in the methods section.

This has been included.

- Line 366: “completely” written twice in the same sentence.

Issue fixed.

- Reorganize the panels of Fig 3.

This figure has been reorganized to simplify it and to bring in the topology diagram from former Fig. S5c. We note, however, that panel 3h may appear to be 'out of order', but we think it's important for clarity that this panel be aligned with panel 3a so that these two sets of data can be easily compared.

- In Fig3d show as atoms R314 and K316 if they are highlighted in the left side of the panel.

We removed the left side of the panel in reorganizing Fig. 3 as it did not add value to the figure as a whole. Atoms R314 and K316 have now been highlighted in Fig. 3d.

- In the legend of Fig5f, a longer description of the proposed mechanism would help the readers.

This legend has been expanded.

- The topology could be added to the alignment in FigS2a.

The topology/secondary structures of the protein have been added above the alignment.

- In FigS7c and S8f the highlight in red of the ATP site could be removed to stress out the mutations pointed.

The ATP coordinating residues highlighted previously in red had been removed from these figures.

Referee #2 (Remarks to the Author):

A. Summary of the key results

The authors identified new (second) activator of CapRel SJ46 bacterial immunity, which binds the same antitoxin domain as the previously discovered activator (and they identified specific residues in CapRel that were involved). They then present multiple crystal structures of the Gp54 ligand alone and in complex with CapRel SJ46, showing a very unusual 6-stranded beta barrel for the Gp54 protein, which dramatically refolds upon CapRel SJ46 binding. They confirmed these structures with specific substitution mutants and HDX mass spectrometry. These structures allowed the authors to create CapRel SJ46 mutants that disrupt Gp54 binding, but not the previously discovered activator MCP.

In Fig 4 the authors return to the Basel collection and testing whether homologs of the two activators, MCP and Gp54, also activated CapRel. One of the more notable findings from this section was the differences between the Gp54 phenotypes from Bas10 and Bas11, which differed only in three amino acids, two of which abolished CapRel SJ46 activation. Because Bas10 and Bas11 had identical MCP but each strain selected for different kinds of escape mutants, the authors then returned to their Bas11 escapers, finding that they only partly escaped immune targeting. Full escape required both MCP and Gp54 mutations, as both were activators of CapRel SJ46. Further, when Gp54 Bas11 was engineered into the other, previously studied phage SECphi27, defense was stronger and no spontaneous mutants were observed.

We thank the reviewer for their careful reading and evaluation of our manuscript, and for their detailed, structured feedback. Below we respond to the individual queries and questions raised.

B. Originality and significance

The authors devote much of their abstract, introduction, and discussion to the fact that CapRel SJ46 can be activated by two distinct phage triggers, and the subsequent implications. While I find this discovery to be interesting and elegant, for me it was not the most surprising part of their paper. I thought that the entropically driven binding involving such extensive protein unfolding and re-folding seemed wild and cool. Admittedly, this is more outside my area of expertise, but if the authors agree with me, they may want to consider refocusing their framing.

So, how unexpected is multifactorial sensing by bacterial defense systems? As the authors state on lines 53-54, for many bacterial immunity proteins, no triggers have been discovered. It is therefore not surprising that no one had previously discovered multiple triggers for a given protein. It is still interesting and cool that the authors have discovered multifactorial sensing here, and very nice that it prevents spontaneous escape mutants from arising, as you would

expect. But I would classify that more as a 'gap in knowledge that is now filled' rather than as a large paradigm shift.

We agree that the entropically driven binding of Gp54 to CapRel^{SJ46} is a fascinating and important discovery. However, we also agree with the other reviewers that the demonstration of multi-factorial sensing is also a critical advance for the anti-phage defense field. As reviewer 3 notes, our work "changes the 1:1 (PRR:PAMP) paradigm currently held in the field of prokaryote immunity". As this reviewer notes, there are not yet that many cases of triggers that have been discovered and appropriately validated. Nevertheless, there are now many instances of phage proteins that have at least been shown to be sufficient to trigger various anti-phage defense systems and a large-scale screen (Stokar-Avihail et al 2022) that found escape mutants, which can often unveil triggers. As summarized in our response to reviewer 1 above about the generalizability of our findings, these prior studies, with the exception of the PARIS work (Burman et al 2024), have implicitly or explicitly assumed that there will be a 1:1 relationship between triggers and defense systems. Thus, we feel our work does represent an important conceptual departure from the current thinking in the field. We have revised the Introduction and Discussion to better capture what's been done previously and how our work significantly differs. That said, we have retained all of the text associated with the entropically driven binding, which remains an important facet of the work that we think further broadens the appeal of the paper. To further emphasize this part of the study, we have added the former Fig. S5c to Fig. 3, which highlights the massive topological change in Gp54 that occurs upon binding CapRel^{SJ46}.

In addition, parallel phenomena have been previously described in animal antiviral immunity. I recommend that in lines 41-48, the authors also discuss and cite eukaryotic immune proteins that can bind different, nonhomologous viral proteins using similar but non-identical molecular interfaces, such as the case of MxA (e.g. doi: 10.1074/jbc.M113.543892, 10.1051/vcm/2022002).

We have added this case to our Introduction and Discussion of eukaryotic immune proteins. As was already noted in the Introduction, the only PRR thought to bind multiple ligands, NAIP/NLRC4, is likely recognizing a common, conserved structural element in those ligands. MxA is an interesting case where divergent nucleoproteins seem to be recognized by different regions of MxA, but there is, to our knowledge, no structural data yet that incisively demonstrates how this is achieved and how the binding interfaces differ. We now also discuss TRIM5 α which binds retroviral capsid proteins, as well as capsid proteins of DNA viruses, although the interaction interface with the capsid proteins of the DNA virus has not been fully mapped.

C. Data & methodology: validity of approach, quality of data, quality of presentation
The approaches and experiments were the right ones to do and the data was of high quality. Most of the figures were well-presented, although I found Fig 3 to be somewhat difficult to decode, which de-emphasized the substantial and cool refolding of Gp54 in the complex. I

suggest that moving the AlphaFold predictions of the MCP complex to the supplement might simplify this figure and make its impact easier to see.

As noted above in response to reviewer 1, we have reorganized Fig. 3 and moved some pieces to the supplement to simplify it, as suggested.

In Fig 5f&g, while I appreciated that the authors were attempting to provide figures that were not photos of spot dilutions, I did not find these figures to be useful. It took a while to decode what the authors were trying to say in these cartoons, and once I figured it out I found it was a message I had already gotten (more clearly) from other parts of the manuscript. Recommend redesigning or dropping these panels.

As noted above, we have expanded the figure legend for Fig. 5f to facilitate the reader's interpretation of that figure panel. We think Fig. 5f-g serve an important role in providing schematic models that summarize the major findings of our work. Fig. 5f is similar to the model presented at the end of our prior study (Zhang et al 2022) and, as noted in the figure legend, Fig. 5g builds off a model from a review by Harmit Malik and colleagues that depicts the traditional 1:1 relationship between coevolving host-pathogen complexes. As our work directly impacts, and elaborates on, this traditional model, we feel it is important to keep in the paper. If there are facets of either Fig. 5f or 5g that the reviewer thinks could be modified for clarity, we would be happy to consider.

D. Appropriate use of statistics and treatment of uncertainties

Many of the datapoints are provided as pictures of spot dilutions and phage plaques, instead of quantification. This is standard in the field and it is nice to see the plaque morphology directly, as this has more information than simple PFU numbers. I do think it would be nice to report how much the plaquing efficiencies varied across the three independent replicates of each experiment that the authors state they performed. Especially in cases where the phenotypes are subtle, I would encourage the authors to report these numbers, perhaps adjacent to the images of the plates or in the supplement if necessary.

There are bar graphs with individual data points shown that quantify and report all of the independent replicates of each plating experiment shown. To keep the main figures relatively simple, these quantifications/graphs are presented as supplementary/extended data figures, with individual numbers provided as source data.

E. Conclusions: robustness, validity, reliability

The evidence reported appears to be strong and is backed up by multiple, independent lines of inquiry. Although see D above.

F. Suggested improvements: experiments, data for possible revision

I do not suggest any additional experiments. My suggestions are limited to those noted elsewhere, specifically:

1) Consider a re-framing of the significance of the manuscript (or re-write these sections to convince readers that multifactorial sensing is as surprising as stated)

Please see our responses to item 'B' above and the revised Introduction.

2) Discuss parallels and/or differences to MxA, and potentially other animal immune genes

Please see our responses to item 'B' above and the revised Introduction and Discussion.

3) Revise the figure layout and presentation in Fig 3 & 5

Figure 3 layout has been substantially revamped. Please see response to item 'C' above regarding Fig. 5.

4) Provide some quantification of the various spot dilution pictures, so that readers can appreciate the variation between replicates.

As noted in response to item 'D', we have ensured that quantifications for all spotting assays are provided in the supplementary/extended data figures and source data.

G. References: appropriate credit to previous work?

The authors are well-versed in the bacterial immunity literature and seem to cite previous work appropriately. I believe they can improve the contextualization of their work with respect to the animal immunity literature.

H. Clarity and context: lucidity of abstract/summary, appropriateness of abstract, introduction and conclusions

-The manuscript is very clear, logical, and well written. The reader is taken along through each step of the data, making it easy to see how and why the conclusions were drawn. I particularly thank the authors for spelling out the conundrum they faced in lines 275-277, which I found very clarifying for the experiments in Fig 5.

-If the authors found co-expression of WT Gp54 and CapRel was toxic in Fig 1, how were they able to co-express them for IPs in Fig 2? I must have missed something here. Would appreciate clarification.

Co-expression of WT Gp54 and CapRel^{S146} is indeed toxic to cells over long periods of time due to translation inhibition. To minimize the effect of toxicity, these experiments were done by only inducing Gp54 for 30 min, which is around/within one generation.

Referee #3 (Remarks to the Author):

Zhang et al. present interesting and timely results that reveal a single prokaryotic PRR can effectively sense two completely different PAMPs. The work is of high significance as it changes the 1:1 (PRR:PAMP) paradigm currently held in the field of prokaryote immunity. The combination of in vivo, structural, and biophysical techniques to thoroughly probe the hypothesis is well executed and strongly supports the conclusions made. The writing and figures are of excellent quality. To do my job as a reviewer well, I scoured the manuscript, supplemental files, and figures to find anything that could improve the manuscript. Although I believe the manuscript is acceptable for publication in its current state, I found two areas that I believe could improve the manuscript – but they are minor.

We thank the reviewer for their comments and for noting the significance of the work to the field.

1. It is clear from the figure legend of figure 3 that the the model of CapRelSJ46 and MCPSECΦ27 was produced by AlphaFold2. However, the text does not indicate this. On lines 170 – 171, and 217-218, the AlphaFold2 model is mentioned in a way that could be interpreted as if it were determined by direct methods. It is recommended that the authors indicate that the model is from an AlphaFold2 prediction in the text in these areas to avoid confusion.

An excellent point - we now explicitly state in the text, at the two places noted, that the MCP-CapRel structure was predicted by AlphaFold. We would add here that this structure was also validated by mutagenesis and HDX, so we think it's a very high confidence prediction.

2. On line 365- 366 there is a statement that reads “Two, dual sensing makes it more difficult for the phage to completely evade defense completely unless both triggers are mutated.” I recommend removing the second “completely” to clarify the meaning of the sentence.

This has been corrected.

Reviewer Reports on the First Revision:

Referees' comments:

Referee #1 (Remarks to the Author):

The authors adequately answered my comments. Congratulations on this great work.

Referee #2 (Remarks to the Author):

My initial comments were minor and mainly focused around the presentation and framing of the results. With the revisions added by the authors to provide additional context to the main text, I am satisfied with the manuscript and have no additional suggestions.